# Limits of Deep Learning: Sequence Modeling through the Lens of Complexity Theory

**Nikola Zubić**[1], **Federico Soldá**[2,*], **Aurelio Sulser**[2,*], **Davide Scaramuzza**[1]

[1]Robotics and Perception Group, University of Zurich, Switzerland
`zubic@ifi.uzh.ch, sdavide@ifi.uzh.ch`
[2]Algorithms and Optimization Group, ETH Zurich, Switzerland
`federico.solda@inf.ethz.ch, asulser@student.ethz.ch`

[*]Equal contribution

## Abstract

Despite their successes, deep learning models struggle with tasks requiring complex reasoning and function composition. We present a theoretical and empirical investigation into the limitations of Structured State Space Models (SSMs) and Transformers in such tasks. We prove that one-layer SSMs cannot efficiently perform function composition over large domains without impractically large state sizes, and even with Chain-of-Thought prompting, they require a number of steps that scale unfavorably with the complexity of the function composition. Also, the language of a finite-precision SSM is within the class of regular languages. Our experiments corroborate these theoretical findings. Evaluating models on tasks including various function composition settings, multi-digit multiplication, dynamic programming, and Einstein's puzzle, we find significant performance degradation even with advanced prompting techniques. Models often resort to shortcuts, leading to compounding errors. These findings highlight fundamental barriers within current deep learning architectures rooted in their computational capacities. We underscore the need for innovative solutions to transcend these constraints and achieve reliable multi-step reasoning and compositional task-solving, which is critical for advancing toward general artificial intelligence.

## 1 Introduction

Deep learning has revolutionized numerous fields, achieving remarkable success in natural language processing (OpenAI, 2023; Google, 2024; Touvron et al., 2023), computer vision (Nguyen et al., 2022; Zubić et al., 2024; Zhu et al., 2024), scientific computing (Merchant et al., 2023; Hansen et al., 2023), and autonomous systems (Kaufmann et al., 2023; Bousmalis et al., 2024). The pursuit of general artificial intelligence now stands as the new frontier, aiming to develop Large Language Models (LLMs) capable of solving novel and complex tasks across diverse domains such as mathematics, coding, vision, medicine, law, and psychology, approaching human-level performance (Bubeck et al., 2023). Mastery of function composition is essential for this objective, as tasks like mathematical problem-solving (Li et al., 2023), learning discrete algorithms (Thomm et al., 2024; Veličković & Blundell, 2021), logical reasoning (Liu et al., 2023b), and dynamic programming (Dziri et al., 2023) are deeply compositional. However, despite impressive capabilities on various language tasks, deep learning models continue to struggle with tasks requiring complex reasoning over sequences, particularly those involving function composition and compositional reasoning (Peng et al., 2024; Dziri et al., 2023).

These tasks necessitate breaking down problems into simpler sub-problems and composing the solutions to these subtasks. Current Transformer models (Vaswani et al., 2017), including advanced ones like GPT-4, find it challenging to handle tasks demanding deep compositionality (Dziri et al., 2023). For instance, we demonstrate that GPT-4 achieves only about 27% accuracy on basic tasks like 4-by-3-digit multiplication. One explanation for this limitation is the Transformer's inability to express simple state-tracking problems (Merrill & Sabharwal, 2023a). Structured State Space Models

(SSMs) (Gu et al., 2022; Gu & Dao, 2023) have been introduced as an alternative to Transformers, aiming to achieve similar expressive power to Recurrent Neural Networks (RNNs) for handling problems that are naturally sequential and require state tracking. While SSMs have demonstrated impressive capabilities on various sequential tasks (Goel et al., 2022; Schiff et al., 2024), they exhibit similar limitations to Transformer models in solving function composition problems. For the same 4-by-3-digit multiplication task, Jamba (Lieber et al., 2024), an SSM-Attention hybrid model, achieves only 17% accuracy.

Existing research has experimentally confirmed the inability of Transformers to perform function composition and compositional tasks (Dziri et al., 2023; Zhao et al., 2024), leading to issues such as hallucinations—responses that are incompatible with training data and prompts. Complexity theory analysis further reveals that Transformers belong to a weak complexity class, logspace-uniform $\mathbf{TC}^0$ (Merrill & Sabharwal, 2023a), as do SSMs (Merrill et al., 2024), emphasizing their inherent limitations. While the impossibility of function composition for Transformers has been theoretically studied (Peng et al., 2024), a similar theoretical understanding for SSMs remains lacking.

In this paper, we address this gap with two main contributions:

1. We provide a theoretical framework using complexity theory to explain the limitations of SSMs in sequence modeling, particularly in their inability to perform function composition efficiently. We prove that one-layer SSMs cannot solve function composition problems over large domains without an impractically large state size (Theorem 1). Additionally, we show that even with Chain-of-Thought prompting, SSMs require a polynomially growing number of steps to solve iterated function composition problems (Theorem 2).

2. We extend our theoretical analysis to multi-layer SSMs, demonstrating that the computation of an $L$-layer SSM on a prompt of length $N$ can be carried out using $O(L \log N)$ bits of memory, positioning SSMs within the complexity class $\mathbf{L}$ (logarithmic space). This implies that SSMs cannot solve problems that are $\mathbf{NL}$-complete unless $\mathbf{L} = \mathbf{NL}$, which is widely believed to be false (Peng et al., 2024). We further discuss that SSMs share this limitation with Transformers, highlighting a fundamental barrier in current deep learning architectures (Theorem 3).

Our critical insight is the formal proof that SSMs cannot solve iterated function composition problems without a polynomially growing number of Chain-of-Thought steps (Theorems 1 and 2) and that even multi-layer finite-precision SSMs are limited to recognizing regular languages due to their essential equivalence to finite-state machines (Theorem 4). While CoT prompting can, to some extent, enable complex problem-solving by breaking down tasks into intermediate steps, it introduces a trade-off between the model's state size and the number of input passes required, leading to increased resource demands, which is not optimal.

These findings underscore the need for innovative solutions beyond current deep learning paradigms to achieve reliable multi-step reasoning and compositional task-solving in practical applications.

## 2 Equivalence of SSMs with Other Deep Learning Models

Recent advancements in deep learning architectures have unveiled significant connections between SSMs and other prevalent models such as Linear Transformers. Notably, Dao & Gu (2024) have demonstrated equivalence between Linear Transformers and SSMs, indicating that the computational processes of these models are fundamentally related. Moreover, SSMs can be trained like Convolutional Neural Networks (CNNs) and inferred as Recurrent Neural Networks (RNNs), leveraging the benefits of both convolutional and recurrent architectures. This duality allows SSMs to efficiently capture long-range dependencies like RNNs while benefiting from the parallelism during training characteristic of CNNs.

Additionally, Merrill et al. (2024) have shown that SSMs and Transformers belong to the same computational complexity class, specifically logspace-uniform $\mathbf{TC}^0$. This alignment in computational capacity reinforces the notion that the limitations observed in SSMs indicate inherent challenges within the broader landscape of deep learning models. Therefore, by focusing our theoretical and empirical analysis on SSMs, we effectively cover the representational capabilities of current deep-learning models, including Transformers and CNNs. This comprehensive coverage justifies our

exploration of the limits of deep learning in sequence modeling through the lens of complexity theory. Our findings highlight the specific shortcomings of SSMs and shed light on the fundamental constraints of deep learning architectures in handling tasks that require reliable multi-step reasoning and compositional task-solving.

## 3 BACKGROUND

For two natural numbers $n \leq m$, we denote $[n] = 1, 2, \ldots, n$ and $[n, m] = n, n + 1, \ldots, m$, with $[0] = [n, n - 1] = \emptyset$. We refer to the number of bits used in each computation as computational precision $p$. Given two domains $B, C$, we denote by $C^B$ the set of all functions from $B$ to $C$.

**Definition 1** (SSM layer). *Given an input sequence $\boldsymbol{x}_1, \ldots, \boldsymbol{x}_n \in \mathbb{R}^m$, an SSM layer $\mathcal{L}$ is defined in terms of a series of matrices $\boldsymbol{A}_t \in \mathbb{R}^{d \times d}$, $\boldsymbol{B}_t \in \mathbb{R}^{d \times m}$, $\boldsymbol{C}_t \in \mathbb{R}^{m \times d}$, and $\boldsymbol{D}_t \in \mathbb{R}^{m \times m}$ for $t \in [n]$. $\mathcal{L}$ defines a sequence of states $\boldsymbol{h}_1, \ldots, \boldsymbol{h}_n \in \mathbb{R}^d$ as*

$$\boldsymbol{h}_t = \boldsymbol{A}_t \boldsymbol{h}_{t-1} + \boldsymbol{B}_t \boldsymbol{x}_t; \tag{1}$$

*and outputs the sequence $\boldsymbol{y}_1, \ldots, \boldsymbol{y}_n \in \mathbb{R}^m$ as*

$$\boldsymbol{y}_t = \boldsymbol{C}_t \boldsymbol{h}_t + \boldsymbol{D}_t \boldsymbol{x}_t. \tag{2}$$

Generally, the matrices $\boldsymbol{A}_t = \boldsymbol{A}(\boldsymbol{x}_t)$, $\boldsymbol{B}_t = \boldsymbol{B}(\boldsymbol{x}_t)$, $\boldsymbol{C}_t = \boldsymbol{C}(\boldsymbol{x}_t)$, and $\boldsymbol{D}_t = \boldsymbol{D}(\boldsymbol{x}_t)$ are functions of the input vector $\boldsymbol{x}_t$ for each $t \in [n]$. In the special case when $\boldsymbol{A}_t, \boldsymbol{B}_t, \boldsymbol{C}_t$, and $\boldsymbol{D}_t$ are independent from the input sequence $\boldsymbol{x}_1, \ldots, \boldsymbol{x}_n$, we call $\mathcal{L}$ a *linear SSM layer*. Moreover, we call $d$ the embedding dimension.

**Remark:** Although SSMs can be linked to streaming algorithms due to their limited hidden state, applying communication complexity to analyze their limitations in function composition involves intricate considerations unique to SSMs. No known streaming lower bound directly applies to our specific setting. Our analysis accounts for the particular architectural constraints of SSMs, providing a better understanding of their capabilities than general streaming algorithms.

## 4 FUNCTION COMPOSITION REQUIRES WIDE ONE-LAYER MODELS

Our analysis considers one-layer SSMs to establish fundamental limitations in function composition tasks. The insights gained at the single-layer level highlight critical challenges that persist even in deeper architectures. The function composition problem has been introduced in (Peng et al., 2024) to provide a theoretical understanding of the causes of the hallucination of Transformer models. The aim is to evaluate the model's capability to combine relational information in the data to understand language, which is the core competence of large language models. Indeed to correctly answer questions like *'what is the birthday of Frédéric Chopin's father?'* given the information that *'the father of Frédéric Chopin was Nicolas Chopin'* and that *'Nicolas Chopin was born on April 15, 1771'*, the model needs to be able to compose the functions *'birthday-of'* and *'father-of'* (Peng et al., 2024), (Guan et al., 2024). Our analysis focuses on function compositions where the functions map elements from one finite, discrete domain to another, such as mapping individuals to their parents or birthdates. These functions operate over discrete sets, like persons and dates, and not over real-valued or continuous domains. Although this function composition task resembles a database join operation, it is important to note that our analysis focuses on how SSMs handle such compositions given natural language prompts. These prompts specify functions in an informal and potentially incomplete manner, lacking the full intensional knowledge present in formal database schemas. We aim to assess the model's ability to perform reasoning over such natural language prompts despite their potential incompleteness.

Next, we give a precise formulation of the *function composition problem* due to (Peng et al., 2024). Consider two functions, $g$ mapping a domain $A$ to a domain $B$, and $f$ mapping $B$ to another domain $C$. These functions will be described in a prompt $X$. The $N$ tokens of $X$ are divided into four parts:

1. the zeroth part describes the argument $x \in A$,
2. the first part describes the function $g$ through $|A|$ sentences in simple, unambiguous language separated by punctuation, e.g., *'the father of Frédéric Chopin is Nicolas Chopin'*,

3. the second part consists of $|B|$ sentences describing the function $f$, e.g., *'the birthday of Nicolas Chopin is April 15, 1771'*,

4. the third part is the query question asking for the value of $f(g(x))$.

In this section, we discuss the theoretical limitations of SSMs for solving the function composition problem. In our analysis, the concept of domain size is crucial. While we primarily consider discrete domains, such as finite sets like $[n] = \{1, 2, \ldots, n\}$, it is important to discuss what domain size means in other contexts. For continuous domains like the interval $[1, n]$, representing general functions would require infinitely many bits, making function composition intractable for models like SSMs and Transformers. Therefore, in practical settings, the maximum meaningful domain size is constrained by the total number of tokens and the prompt length, as the model's input capacity is limited. In our composition tasks, the functions are described within the prompt, so the prompt length effectively serves as an upper bound on the domain size.

**Theorem 1.** *Consider a function composition problem with input domain size $|A| = |B| = n$ and an SSM layer $\mathcal{L}$ with embedding dimension $d$ and computation precision $p$. Let $R = n \log n - (d^2 + d)p \geq 0$, then the probability that $\mathcal{L}$ answers the query incorrectly is at least $R/(3n \log n)$ if $f$ is sampled uniformly at random from $C^B$.*

The proof is based on a reduction from a famous problem in communication complexity (Peng et al., 2024), (Yao, 1979). Additional background on Communication Complexity and relevant problem classes can be seen in the Appendix A. We have three agents dubbed Faye, Grace, and Xavier. We assume that the agents have unbounded computational capabilities but, the only communication allowed is from Faye and Grace to Xavier. Faye knows a function $f : [n] \mapsto [n]$ and the argument $x \in [n]$, Grace knows a function $g : [n] \mapsto [n]$ and the argument $x$, while Xavier only knows the argument $x \in [n]$. The goal is for Xavier to compute the value of $f(g(x))$, minimizing the total number of bits communicated from Faye to Xavier and from Grace to Xavier.

We report a lemma from (Peng et al., 2024), which gives a hardness result for the abovementioned problem.

**Lemma 1** (Lemma 1 from (Peng et al., 2024)). *Consider the problem described above: if fewer than $n \log n$ bits are communicated by Faye to Xavier, then Xavier cannot know the value $f(g(x))$. In particular, if only $n \log n - R$ bits are communicated for some $R \geq 0$, then the probability that the composition is computed incorrectly is at least $R/(3n \log n)$ if $f$ is sampled uniformly at random from $C^B$.*

Now, we prove the theorem based on the Lemma above.

*Proof of Theorem 1.* To establish the bound on $q$, we give a reduction of the communication problem above to the function composition problem. Let $\mathcal{L}$ be an SSM layer that can solve the function composition problem with probability $q$.

Suppose we have Faye, Grace, and Xavier as in the settings above, and Xavier wants to find the value $f(g(x))$. We construct the following prompt: the zeroth token $\boldsymbol{x}_0$ is *'the argument of the function is $x$'*, for $i \in [1, n]$ let $\boldsymbol{x}_i$ be the token *'g applied to $i$ is $g(i)$'*, where the information is provided by Grace, and for $i \in [n+1, 2n]$ let $\boldsymbol{x}_i$ be the token string *'f applied to $i$ is $f(i)$'*, where the information is provided by Faye. Xavier provides the last token string $\boldsymbol{x}_{2n+1} = $ *'what is the value of $f(g(x))$'*. Since the SSM layer $\mathcal{L}$ can solve the composition task with probability $q$, we have that:

$$\boldsymbol{y}_{2n+1} = \boldsymbol{C}_{2n+1}\boldsymbol{h}_{2n+1} + \boldsymbol{D}_{2n+1}\boldsymbol{x}_{2n+1} = f(g(x)) \tag{3}$$

with probability $q$.

But this allows us to construct the following communication protocol. Since Grace knows $g$ and the argument $x$, she knows the values of $\boldsymbol{x}_i$ for $i \in [0, n]$ and she iteratively computes:

$$\boldsymbol{h}_i = \boldsymbol{A}_i\boldsymbol{h}_{i-1} + \boldsymbol{B}\boldsymbol{x}_i, \tag{4}$$

and then sends $\boldsymbol{h}_n$ to Xavier. On the other hand, Faye knows $f$ and hence the values of $\boldsymbol{x}_i$ for $i \in [n+1, 2n]$, she computes the matrix:

$$\mathcal{A} = \prod_{j=n+1}^{2n} \boldsymbol{A}_j, \tag{5}$$

then the vector:

$$\boldsymbol{b} = \sum_{i=n+1}^{2n} \left( \prod_{j=n+1}^{2n-i} \boldsymbol{A}_j \right) \boldsymbol{B}_i \boldsymbol{x}_i, \tag{6}$$

and she sends them to Xavier. At this point, Xavier computes:

$$\boldsymbol{h}_{2n+1} = \boldsymbol{A}_{2n+1} \cdot (\mathcal{A} \cdot \boldsymbol{h}_n + \boldsymbol{b}) + \boldsymbol{B}_{2n+1} \boldsymbol{x}_{2n+1}. \tag{7}$$

and finds the value of $f(g(x))$ with probability $q$ by computing $\boldsymbol{y}_{2n+1} = \boldsymbol{C}_{2n+1} \cdot \boldsymbol{h}_{2n} + \boldsymbol{D}_{2n+1} \cdot \boldsymbol{x}_{2n+1}$. The total number of bits of communication between Faye and Xavier is $(d^2 + d) \cdot p$. By Lemma 1, it follows that $q \leq R/(3n \log n)$. □

Our theoretical results in Theorem 1 highlight that SSMs, like other deep neural networks, approximate functions rather than perform symbolic reasoning. Specifically, the probability bound indicates that if we attempt to compose functions over domains of size $n$ with an SSM of embedding dimension $d$ and computational precision $p$ such that $(d^2 + d)p < n \log n/2$, the model will output the incorrect result with a probability of at least $1/6$. To achieve a high probability of correctness (e.g., 99%), $(d^2 + d)p$ must be significantly larger than $n \log n/2$. This establishes a strong lower bound on the model's width, demonstrating that to accurately perform function composition over large domains, the model's capacity must increase substantially.

While Theorem 1 addresses the limitations of one-layer SSMs, a natural question arises: Can deeper SSMs overcome these limitations? We conjecture that any SSM with a constant number of layers would still be unable to resolve the iterated composition task (as formalized in our Chain-of-Thought section 5). This is because accurately communicating token embeddings between layers becomes increasingly challenging as the depth grows. The difficulty in preserving and transmitting the necessary information across layers suggests that simply increasing the number of layers without a corresponding increase in model capacity does not address the fundamental limitations identified.

## 5 MANY THOUGHT STEPS ARE NEEDED

A chain of thought (CoT) is a series of intermediate natural language reasoning steps that lead to the final output. In this section, we focus on language models that can generate a similar chain of thought— a coherent series of intermediate reasoning steps that lead to the final answer for a problem. In (Wei et al., 2022), it was observed that CoT can mitigate the issue of hallucinations by encouraging the LLM to generate prompts that break down the task into smaller steps, eventually leading to the correct answer. In this section, we prove that, in general, many CoT steps are needed to break down compositional tasks.

We start the discussion with the formal definition of an SSM with $k$ CoT steps. It adapts the definition for the Transformer model of (Merrill & Sabharwal, 2024) to the case of SSMs.

**Definition 2** (SSM with CoT). *Let $\phi : (\mathbb{R}^m)^* \to \mathbb{R}^m$ be a function mapping a prefix of tokens to a new token. The function $\phi$ is parametrized by an SSM layer $\mathcal{L}$.*

*Given an input sequence $\boldsymbol{x}_1, \boldsymbol{x}_2, \ldots, \boldsymbol{x}_n \in \mathbb{R}^m$, we call:*

$$\phi_k(\boldsymbol{x}_1, \boldsymbol{x}_2, \ldots, \boldsymbol{x}_n) = \phi_{k-1}(\boldsymbol{x}_1, \boldsymbol{x}_2, \ldots, \boldsymbol{x}_n) \cdot \phi(\phi_{k-1}(\boldsymbol{x}_1, \boldsymbol{x}_2, \ldots, \boldsymbol{x}_n), \boldsymbol{x}_1, \boldsymbol{x}_2, \ldots, \boldsymbol{x}_n),$$

*where $\phi_1(\boldsymbol{x}_1, \boldsymbol{x}_2, \ldots, \boldsymbol{x}_n) = \phi(\boldsymbol{x}_1, \boldsymbol{x}_2, \ldots, \boldsymbol{x}_n)$ and $\cdot$ denotes concatenation, the output of the SSM layer $\mathcal{L}$ with $k$ CoT-steps.*

In this section, we want to prove that while this procedure could help SSM layers with compositional tasks, it might require many CoT steps to be effective. In particular, we focus on the iterated function composition problem and show a lower bound on the number of CoT steps needed by an SSM layer to solve this problem correctly.

In the *iterated function composition* problem we are given $k$ functions $f_1, f_2, \ldots, f_k : [n] \mapsto [n]$, and we need to calculate $f_k(f_{k-1}(\ldots f_2(f_1(x)) \ldots))$ for $x \in [n]$. Here we restrict to the case when $f_1 = f_2 = \cdots = f_k$, we define $f^{(k)}(x) := f(f(\ldots f(x)))$, and we call this $k$-*iterated function composition* problem.

**Theorem 2.** *Consider an iterated composition problem with domain size $n$, computation precision $p$, and embedding dimension $d$. An SSM layer requires $\Omega(\frac{\sqrt{n \log n}}{dp})$ CoT steps for answering correctly iterated function composition prompts.*

The proof relies on reducing the iterated function composition problem from the pointer chasing problem (Papadimitriou & Sipser, 1982), a classical problem in communication complexity. In the *k-steps pointer chasing* problem, we have two agents dubbed Alice and Bob; Alice knows a function $f_A : [n] \mapsto [n]$ and Bob knows a function $f_B : [n] \mapsto [n]$. We then define the pointers:

$$z_1 = 1, \quad z_2 = f_A(z_1), \quad z_3 = f_B(z_2), \quad z_4 = f_A(z_3), \quad z_5 = f_B(z_4), \quad \dots.$$

The communication proceeds for $2k$ rounds, with Alice starting. The goal is for Bob to output the binary value of $z_{2k+2} \mod 2$. Following, we prove that an SSM layer with $R$ CoT steps solving the iterated function composition problem can be used to design a communication protocol for the pointer chasing problem where the number of transmitted bits scales with $R$. The next fundamental Lemma in communication complexity gives a lower bound on the number of bits that need to be communicated in any such communication protocol, thus allowing the lower bound to be derived on the CoT steps.

**Lemma 2** (Theorem 1.1 (Yehudayoff, 2020)). *Any randomized protocol for the k-steps pointer chasing problem with error probability $1/3$ under the uniform distribution must involve the transmission of at least $n/(2000k) - 2k \log n$ bits.*

Before we begin with the actual proof, let us introduce some notation. We note that $\phi_k$ is a string of $k$ tokens of $\mathbb{R}^m$. Moreover, to compute the new token $\phi(\phi_{k-1}(\boldsymbol{x}_1, \boldsymbol{x}_2, \dots, \boldsymbol{x}_n), \boldsymbol{x}_1, \boldsymbol{x}_2, \dots, \boldsymbol{x}_n)$ the SSM layer $\mathcal{L}$ computes $n + (k-1)$ hidden states. We denote the $i$-th hidden state by $\phi_{k,i}(\boldsymbol{x}_1, \dots, \boldsymbol{x}_n)$.

*Proof of Theorem 2.* The proof is similar to the proof of Theorem 2 in (Peng et al., 2024). We reduce the pointer chasing problem to the iterated composition problem with CoT prompts. In particular, we show that if the SSM $\mathcal{L}$ can solve the $k$-iterated function composition problem with $R$ CoT steps, then we can construct a protocol solving the $(k-1)$-steps pointer chasing problem using $2Rdp$ bits of communication.

Fix a $(k-1)$-steps pointer chasing problem for the function $f_A, f_B : [n] \mapsto [n]$. Define the function $f : [2n] \mapsto [2n]$ as:

$$f(i) = \begin{cases} f_A(i) + n, & i \in [1, n]; \\ f_B(i - n), & i \in [n+1, 2n]. \end{cases} \tag{8}$$

We point out that $f^{(k)}(i) = (f_B \circ f_A)^{(k)}(i)$. Consider the $k$-iterated function composition problem for $f$ and suppose that there exists an SSM $\mathcal{L}$ that solves it using $R$ CoT steps.

We construct the following prompt: for $i \in [1, n]$ let $\boldsymbol{x}_i$ be the token '*f applied to i is $f(i)$*', where the information $f(i)$ is provided by Alice, and for $i \in [n+1, 2n]$ let $\boldsymbol{x}_i$ the token string '*f applied to i is $f(i)$*', where the information $f(i)$ is provided by Bob. The last token string $\boldsymbol{x}_{2n+1}$ is given by '*what is the value of $f^{(k)}$ applied to x*'. Since the SSM layer $\mathcal{L}$ can solve the $k$-iterated function composition task with $R$ CoT steps, we have that $\phi_R(\boldsymbol{x}_1, \dots, \boldsymbol{x}_{2n})$ is the right answer for $f^k(x)$. We will use this fact to construct a communication protocol transmitting at most $2 \cdot Rdp$ bits. The communication protocol lasts for $R$ rounds.

In the $r$-th round Alice computes $\phi_{r,n+k}(\boldsymbol{x}_1, \dots, \boldsymbol{x}_{2n})$ from $\phi_{r-1}(\boldsymbol{x}_1, \dots, \boldsymbol{x}_{2n})$ (where $\phi_0(\boldsymbol{x}_1, \dots, \boldsymbol{x}_{2n})$ is the empty string of tokens) and $\boldsymbol{x}_1, \dots, \boldsymbol{x}_n$ and communicates it with Bob. Bob on the other hand computes $\phi_r(\boldsymbol{x}_1, \dots, \boldsymbol{x}_{2n})$ from $\phi_{r,n+k}(\boldsymbol{x}_1, \dots, \boldsymbol{x}_{2n})$ and $\boldsymbol{x}_{n+1}, \dots, \boldsymbol{x}_{2n}$ and transmits it to Alice. In each iteration, at most $dp$ bits are communicated from Alice to Bob and from Bob to Alice.

After $R$ rounds, Bob knows the value of $\phi_R(\boldsymbol{x}_1, \dots, \boldsymbol{x}_{2n})$. By hypothesis, this is the solution to the $(k-1)$-steps pointer chasing problem. Notice that the total number of bits communicated by the protocol is $2Rdp$. In conclusion, we fix $k = \frac{1}{100}\sqrt{\frac{n}{\log n}} + 1$ and by Lemma 2 we get that $2Rdp \geq n/(2000k) - 2k \log n$ which gives $R \geq \frac{3}{100}\frac{\sqrt{n \log n}}{dp}$ $\qquad \square$

# 6  SSMs Are Limited to Regular Languages

In Peng et al. (2024), it is suggested to analyze the computational capability of LLMs on the computational problems below. The empirical compositional tasks studied in later sections—multiplication of multi-digit integers, dynamic programming, and logic puzzles such as "Einstein's Riddle"—can be expressed in terms of these computational problems (Peng et al., 2024).

**Circuit evaluation**: Given the description of a circuit with gates, which can be either Boolean or arithmetic operations, as well as the values of all input gates of the circuit, evaluate the output(s) of the circuit. Multiplying decimal integers with multiple digits is an example of such a circuit.

**Derivability**: Given a finite domain $S$ and a relation $D \subseteq S \times S$. For a given initial set $I \subseteq S$ and a final set $F \subseteq S$, answer the question whether there are elements $a_1, a_2, \ldots, a_k \in S$ such that (a) $a_0 \in I$, (b) $a_k \in F$, and (c) for all $j$ such that $0 < j \leq k$, $(a_{j-1}, a_j) \in D$.

**Logical reasoning**: Logic puzzles like 'Einstein's Riddle' can typically be formulated as satisfiability (or SAT) instances. This problem is NP-complete. However, most common-sense reasoning can be expressed by one of the three tractable exceptional cases of SAT: 2-SAT, Horn SAT, and Mod 2 SAT.

In Peng et al. (2024), it was noted that Derivability and 2-SAT are **NL**-complete, while Horn SAT and Circuit Evaluation are **P**-complete problems. Since the log-precision Transformer model lies in the complexity class log-uniform $\mathbf{TC}^0 \subseteq \mathbf{L}$ (Merrill & Sabharwal, 2023b), these problems cannot be solved by a log-precision Transformer model provided $\mathbf{NL} \neq \mathbf{L}$ (which is a widely believed hypothesis in computational complexity). For Mod 2 SAT, the result is valid provided the weaker statement $\mathbf{L} \neq \text{Mod } 2\,\mathbf{L}$. For Horn SAT and Circuit Evaluation, the result holds unless the stronger statement $\mathbf{L} = \mathbf{P}$ holds. Very recently, in Merrill et al. (2024), it was established that log-precision linear and S6-SSMs (Gu & Dao, 2023) are also part of the complexity class log-uniform $\mathbf{TC}^0$, which yields the following theorem similar to the case of Transformers.

**Theorem 3.** *The problems of Derivability and 2-SAT cannot be solved by log-precision linear or S6-SSMs provided $\mathbf{L} \neq \mathbf{NL}$. For Mod 2 SAT, the result is valid provided the weaker statement $\mathbf{L} \neq \text{Mod } 2\,\mathbf{L}$ holds. For Horn SAT and Circuit Evaluation, the result holds unless the stronger statement $\mathbf{L} = \mathbf{P}$ holds.*

So far, we have explored SSMs' computational capabilities and limitations in various settings. In particular, SSMs face fundamental challenges when handling tasks requiring complex reasoning or computations beyond their inherent architectural constraints. Building on these insights, we now examine the impact of finite precision arithmetic on the computational power of SSMs. In practical implementations, SSMs operate with finite precision due to hardware constraints, typically using fixed-point or floating-point representations with a limited number of bits. This finite precision restricts the range and granularity of values that the model's parameters and hidden states can represent. Let us denote by $\mathcal{S}$ the vectors that the hidden states can take. Consequently, it is reasonable to assume that an SSM does not embed or store all the information provided during training. Instead, it must extract and retain only a subset of relevant information within a memory of much smaller capacity than the input.

Since SSMs have a fixed hidden dimension $d$ and operate with finite precision, the set $\mathcal{S}$ is finite. This finiteness imposes significant restrictions on the types of languages that SSMs can compute or recognize. To formalize this limitation, we introduce the following definition:

**Definition 3.** *Given a finite-precision SSM, we say the SSM accepts an input sequence $\boldsymbol{x}_1, \ldots, \boldsymbol{x}_n$ if the hidden state $\boldsymbol{h}_{n+1} \neq \boldsymbol{0}$. We call the set of accepted input sequences the language of the SSM.*

The following theorem relates the computational power of a finite precision SSM to a Finite-State Machine (FSM).

**Theorem 4.** *The language of a finite-precision SSM is within the class of regular languages.*

*Proof.* We construct the Finite-State Machine $M = (Q, \Sigma, \delta, q_0, F)$, where:

1. the set of states $Q$ is the set of all possible vectors $\int$,

2. the finite input alphabet $\Sigma$ is as well $\mathcal{S}$,

3. the transition function $\delta : Q \times \Sigma^2 \to Q$ is defined by the SSM's state update equations, i.e., $\delta(\boldsymbol{h}, \boldsymbol{x}) = \boldsymbol{A}(\boldsymbol{x})\boldsymbol{h} + \boldsymbol{B}(\boldsymbol{x})\boldsymbol{x}$

4. $q_0$ is the initial state corresponding to the initial hidden state $\boldsymbol{h}_0$,

5. $F = Q \setminus \{\boldsymbol{0}\}$ is the set of accepting states.

Since the SSM has finite precision, the set $\mathcal{V}$ is finite, and this defines a valid FSM. It is immediate that the FSM models the transitions of the SSM and accepts the same language. □

This result relates the computational power of SSMs to FSMs. Consequently, under finite precision constraints, SSMs cannot recognize languages beyond regular languages. Regarding computational limitations, tasks requiring computational models with greater expressive power, such as context-free grammars or context-sensitive grammars, cannot be **efficiently solved** by SSMs with finite precision. Examples of such tasks include recognizing balanced parentheses, detecting palindromic sequences, and performing more complex logical inference that necessitates memory beyond finite states.

These limitations are significant because they highlight the boundaries of what SSMs can achieve in practical settings. Regarding practical considerations, since real-world implementations of SSMs operate on hardware with finite memory and finite precision arithmetic, these theoretical limitations directly apply to SSMs used in actual applications. Therefore, when designing systems for tasks that require processing beyond regular languages, it becomes clear that SSMs with finite precision do not suffice, and alternative architectures or computational mechanisms need to be considered to overcome these inherent constraints.

## 7 EXPERIMENTS

Our theoretical results suggest that SSMs inherently struggle with function composition and multi-step reasoning tasks due to their architectural limitations. To validate these findings, we empirically assess SSMs' performance on practical tasks requiring these capabilities.

We evaluate the inability of various sequence models to address function composition tasks by examining three axes of composition: spatial, temporal, and relational (Appendix B.1). This evaluation uses four datasets designed to test function composition. Subsequently, we proceed to compositional tasks involving multi-digit multiplication, dynamic programming, and Einstein's puzzle. We investigate the effects of Chain-of-Thought (CoT) prompting (Appendix B.2) and conduct a thorough error analysis to understand the failure points and underlying reasons for the erroneous behavior (Appendix B.3).

We conducted GPT experiments using the ChatGPT API (OpenAI, 2023) and performed all experiments with the GPT-4 model as of June 2024, while other models were evaluated on machines equipped with 2x NVIDIA A100 80 GB GPUs. We used Jamba version 1. Unless otherwise specified, each task is evaluated three times with 500 test samples per evaluation to ensure consistency and minimize variance. All other experimental details, including prompts and additional results, are provided in the Appendix.

## 8 RELATED WORK

**Limitations in Function Composition and Reasoning** Recent studies have underscored the limitations of deep learning models, particularly Transformers, in handling tasks requiring deep compositionality and multi-step reasoning (Peng et al., 2024; Dziri et al., 2023). These tasks are crucial in applications like mathematical problem-solving (Li et al., 2023), algorithm learning (Thomm et al., 2024; Veličković & Blundell, 2021), logical reasoning (Liu et al., 2023b), and dynamic programming (Dziri et al., 2023). Despite their capabilities, Transformers have been shown to struggle with function composition, which is essential for understanding relational information in data (Guan et al., 2024).

Research has highlighted architectural and training limitations that prevent these models from maintaining accuracy over multiple reasoning steps, leading to issues like hallucinations and reasoning errors (Merrill & Sabharwal, 2023a; Zhao et al., 2023). Studies by Merrill et al. (2024) and Peng et al.

(2024) have identified that both Transformers and SSMs belong to weak complexity classes, such as logspace-uniform $\mathbf{TC}^0$, which limits their computational abilities. However, prior work primarily focused on Transformers, with SSMs not thoroughly investigated theoretically and empirically concerning their ability to perform function composition and compositional tasks. Our contribution fills this gap by providing a comprehensive theoretical framework and empirical analysis specific to SSMs.

**Chain-of-Thought Prompting**  The Chain-of-Thought (CoT) prompting method has been proposed to improve reasoning capabilities in large language models by breaking down complex tasks into smaller, intermediate steps (Wei et al., 2022). CoT prompting aims to mitigate issues like hallucinations and enhance multi-step reasoning by encouraging models to generate intermediate reasoning steps. While CoT has shown promise in specific contexts, recent research indicates that even with CoT prompting, current models remain inadequate for solving deeply compositional tasks (Merrill & Sabharwal, 2023a; Liu et al., 2023a). Our work supports these findings, demonstrating that CoT prompting does not overcome the fundamental computational limitations of SSMs and Transformers in tasks requiring complex reasoning.

While advanced methods like tree search algorithms (Trinh et al., 2024; Polu & Sutskever, 2020; Lample et al., 2022) and self-correction techniques (Wang et al., 2024; Kumar et al., 2024) have been proposed to improve reasoning by integrating external mechanisms, our work focuses on the inherent computational limitations of SSMs and Transformers when used without such augmentations. These external engines can mitigate some limitations by leveraging additional resources, but they do not address the core architectural constraints we have identified.

**Expressive Power and Complexity of Neural Networks**  A growing body of work is exploring the expressive power of neural network architectures and their limitations from a computational complexity perspective. Weiss et al. (2018) and Siegelmann & Sontag (1992) examined the capabilities of recurrent neural networks in relation to Turing machines. Pérez et al. (2019) investigated the Turing completeness of Transformers under certain conditions.

More recently, Merrill et al. (2020) analyzed the relationship between network depth, parameter size, and computational expressivity. Bhattamishra et al. (2020) explored the computational limitations of Transformers concerning formal languages. Our work contributes to this line of research by analyzing SSMs within the framework of computational complexity, specifically their placement within the class $\mathbf{L}$ and implications for their reasoning capabilities.

**Alternative Approaches to Complex Reasoning**  Given the limitations of current architectures, researchers have explored alternative approaches to enhance models' reasoning abilities. Methods include integrating external memory modules (Graves et al., 2016), incorporating symbolic reasoning components (Gaunt et al., 2017), and developing neuro-symbolic models (Dai et al., 2019). These approaches aim to combine the strengths of neural networks with symbolic computation to overcome the shortcomings in tasks requiring complex, multi-step reasoning. Our findings underscore the necessity for such innovative solutions, suggesting that overcoming the fundamental limitations identified requires moving beyond traditional deep learning paradigms.

## 9  CONCLUSION

In this work, we have demonstrated both theoretically and empirically that Structured State Space Models (SSMs) and Transformers face fundamental limitations in performing function composition and complex reasoning tasks. Our theoretical analysis shows that overcoming these limitations would require architectures beyond finite-state machines. SSMs with fixed hidden dimensions and layers are equivalent to finite-state machines and thus limited to regular languages (Theorem 4). This limitation explains their inability to handle tasks that require computational power beyond regular languages, such as context-free languages or problems that are $\mathbf{NL}$-complete.

Our empirical evaluations confirm these findings, revealing significant performance degradation as task complexity increases, even when employing advanced prompting techniques. Models often resort to shortcuts, leading to errors in multi-step reasoning processes. These results highlight that current deep learning architectures are fundamentally limited in their ability to perform reliable

multi-step reasoning and compositional task-solving due to their architectural constraints. This underscores the necessity for innovative architectural solutions or computational frameworks that can handle such tasks more efficiently. Future research should explore new directions, such as integrating symbolic reasoning components, improving memory and state-tracking capabilities, or developing hybrid models that transcend the limitations of existing architectures. Addressing these challenges is crucial for advancing toward general artificial intelligence capable of sophisticated reasoning and problem-solving across diverse domains.

## 10 ACKNOWLEDGMENT

This work was supported by the European Research Council (ERC) under grant agreement No. 864042 (AGILEFLIGHT).

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

# Appendices

# A BACKGROUND ON COMMUNICATION COMPLEXITY AND COMPUTATIONAL CLASSES

To provide a solid foundation for our theoretical results, we offer an overview of key concepts in communication complexity and computational complexity theory. This background is essential for understanding the limitations of SSMs and other deep learning architectures in sequence modeling tasks that require complex reasoning.

## A.1 COMMUNICATION COMPLEXITY

**Communication complexity** studies the amount of communication required between two or more parties to compute a function whose input is distributed among them. It provides lower bounds on the communication needed for distributed computation.

**Communication Protocols:** A communication protocol specifies the rules by which parties exchange messages to compute a function collaboratively. The primary goal is to minimize the total number of bits exchanged.

**Key Problems in Communication Complexity:**

- *Function Composition Problem:* Two parties, **Faye** and **Grace**, hold functions $f : B \to C$ and $g : A \to B$, respectively, along with a common input $x \in A$. Their goal is to compute $f(g(x))$ with minimal communication to a third party, **Xavier**. This problem models scenarios where composing functions over large domains requires significant communication, highlighting the challenges in function composition tasks for sequence models.

- *Pointer Chasing Problem:* This involves two parties who alternately apply functions to an initial input over several rounds. It is a fundamental problem used to establish lower bounds in communication complexity. It demonstrates that certain computations inherently require a substantial amount of communication, regardless of the protocol used.

- *Set Disjointness Problem:* Two parties each hold a subset of a universal set and wish to determine if their subsets intersect without revealing additional information. This problem is notable for having high communication complexity, serving as a basis for proving lower bounds in various computational models.

**Relevance to Sequence Modeling:** Communication complexity provides tools to prove theoretical limits on the capabilities of computational models, including neural networks. By reducing problems in communication complexity to tasks performed by sequence models, we can establish lower bounds on the resources required (e.g., hidden state size, number of layers) for these models to perform certain computations. This approach helps in understanding why models like SSMs struggle with tasks requiring complex reasoning or function composition.

## A.2 COMPUTATIONAL COMPLEXITY CLASSES

Computational complexity theory classifies problems based on the resources required to solve them, such as time or memory space. Understanding these classes is crucial for characterizing the limitations of computational models.

**Key Complexity Classes:**

- **L (Logarithmic Space):** The class of decision problems solvable by a deterministic Turing machine using logarithmic amount of memory space with respect to the input size. Problems in L are considered efficiently solvable with very limited memory.

- **NL (Nondeterministic Logarithmic Space):** Consists of decision problems solvable by a nondeterministic Turing machine using logarithmic space. NL is a broader class than L, as nondeterminism allows guessing and verifying solutions using limited memory.

- **P (Polynomial Time):** Contains decision problems solvable by a deterministic Turing machine in polynomial time. It represents problems that are efficiently solvable in terms of time, without specific memory constraints.

- **Regular Languages:** The class of languages recognizable by finite automata or equivalently, by regular expressions. They are the simplest class in the Chomsky hierarchy and can be recognized using constant memory.

- **Context-Free Languages:** Recognizable by pushdown automata, these languages can handle nested structures and require memory that grows with the input size.

- **$TC^0$ (Constant Depth Threshold Circuits):** A class of problems solvable by constant-depth, polynomial-size circuits with threshold gates. These circuits can compute certain functions very efficiently in parallel.

**Relationships Between Classes:**

$$\text{Regular Languages} \subseteq \mathbf{L} \subseteq \mathbf{NL} \subseteq \mathbf{P} \tag{9}$$

It's widely believed that these inclusions are strict (e.g., $\mathbf{L} \neq \mathbf{NL}$), meaning each class strictly contains the previous one.

**Relevance to Sequence Modeling:** By placing computational models within these complexity classes, we can formalize their computational power and limitations. For instance:

- **Finite-State Machines (FSMs):** Equivalent to models that recognize regular languages. They have a finite number of states and cannot handle tasks requiring memory that scales with input size.

- **Pushdown Automata:** Recognize context-free languages and can handle nested or recursive structures due to their use of a stack.

- **SSMs and Transformers:** Our analysis shows that SSMs with fixed hidden dimensions and layers are equivalent to FSMs, limiting them to regular languages. Similarly, Transformers have been shown to have limitations corresponding to the class $TC^0$ or L under certain conditions.

**Implications for SSMs:** Understanding that SSMs are limited to regular languages explains why they struggle with tasks requiring more computational power, such as:

- *Function Composition:* Requires the ability to maintain and manipulate information over long sequences, which exceeds the capabilities of finite-state models.

- *Complex Reasoning Tasks:* Problems like multi-digit multiplication, logical puzzles, and dynamic programming necessitate memory and computational resources beyond what is available in models limited to regular languages.

By grounding our analysis in communication complexity and computational complexity theory, we establish a theoretical foundation for the limitations of SSMs. This background enables us to formalize the challenges faced by sequence models in handling tasks that require computational resources beyond regular languages and logarithmic space.

A.3 Key Problems and Their Complexity

To further contextualize the limitations of SSMs, we briefly describe some computational problems and their associated complexity classes:

- **Derivability (NL-Complete):** Given a finite set and a relation, determine if there is a sequence of elements satisfying certain conditions. This problem requires nondeterministic logarithmic space and cannot be solved by models limited to L unless $\mathbf{L} = \mathbf{NL}$.

- **2-SAT (NL-Complete):** A satisfiability problem where each clause has at most two literals. It is solvable in polynomial time but is NL-complete, indicating it requires more than deterministic logarithmic space.

- **Horn SAT and Circuit Evaluation (P-Complete):** Problems that are as hard as any problem in P. Solving these efficiently would require polynomial time computation, beyond the capabilities of FSMs.

- **Mod 2 SAT (Beyond L):** Involves solving satisfiability problems modulo 2. Requires computational resources beyond deterministic logarithmic space.

**Relevance to Our Work:** The inability of SSMs to solve these problems stems from their equivalence to finite-state machines. Since FSMs cannot utilize memory that grows with input size, they are inherently incapable of solving problems that require maintaining and processing an unbounded amount of information. The limitations highlighted by these complexity classes and problems suggest that to handle complex reasoning tasks effectively, sequence models need architectures that go beyond finite-state computations. This could involve models that can simulate pushdown automata or Turing machines, allowing them to recognize context-free languages or perform computations requiring more substantial memory resources.

# B   MAIN EXPERIMENTS

## B.1   FUNCTION COMPOSITION AND COMPOSITIONAL TASKS

In the context of Large Language Models (LLMs), compositional tasks differ from function composition. Function composition $f_K(f_{K-1}(\dots(f_1(x))))$ is a mathematical process where the output of one function serves as the input for another across multiple functions $f_1, f_2, \dots, f_K$. Conversely, LLM compositional tasks involve breaking down complex inputs into simpler parts and integrating the results to generate an overall output. Examples include (i) combining linguistic elements to generate coherent text, (ii) solving multi-step reasoning problems, and (iii) decomposing complex tasks (e.g., multi-turn conversations, summarization) into manageable sub-tasks.

Solving compositional tasks necessitates the capability to perform function composition (Peng et al., 2024; Dziri et al., 2023) and demands additional competencies such as contextual understanding, multi-step reasoning, and the integration of diverse information types. A model's proficiency in function composition is a critical prerequisite for tackling complex compositional tasks (Lu et al., 2023). For instance, if an SSM-powered LLM cannot evaluate $f(g(x))$, it will be inadequate for tasks involving multi-step arithmetic or logical operations that depend on nested functions.

**Composition tasks**   We begin with three fundamental composition tasks: spatial, temporal, and relationship compositions. These axes are crucial as they encapsulate core aspects of comprehending and interacting with the world. Spatial composition entails integrating information about the positions and orientations of objects. Temporal composition involves reasoning over sequences and durations of events. Relationship composition focuses on understanding the connections between entities, such as those in a genealogy tree.

**Number of Parameters**   We conducted experiments using Jamba (Lieber et al., 2024) (joint Mamba and Attention) with 7B parameters, Mamba (Gu & Dao, 2023) with 2.8B parameters, S4-H3 (Gu et al., 2022; Fu et al., 2023) with 2.7B parameters, GPT-4 (OpenAI, 2023), and GPT-4o models. Qualitative results are presented in Fig. 1. As illustrated in Fig. 1, all models failed to answer questions across the three composition axes correctly.

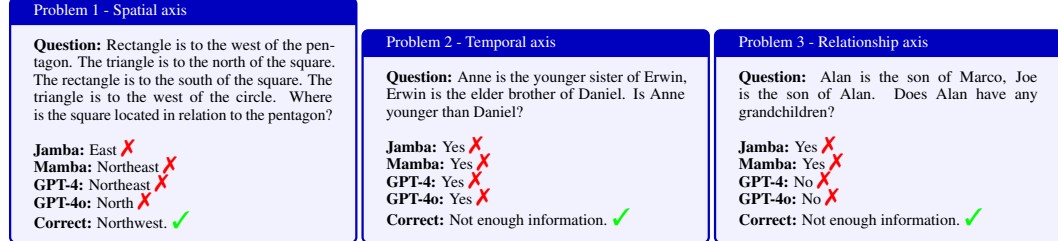

Figure 1: Qualitative example of zero-shot inference on prominent SSM and Attention-based models. None of the models successfully resolved the problems across any of the composition axes.

To quantitatively assess the limitations of models, including the latest GPT-4o (OpenAI, 2023), in solving function composition tasks, we evaluate their performance on four datasets specifically designed to test these capabilities. Unless otherwise specified, each model is tested on 500 samples.

**Composition datasets** *Math-QA* dataset, derived from (Li et al., 2023), includes 25 math topics. We focus on the first 100 samples from Algebra, Calculus, Combinatorics, Game Theory, and Trigonometry. Problems involve solving function compositions and temporal reasoning. ***BIG-Bench Hard*** (Suzgun et al., 2022) dataset features 250 Boolean expressions that the model must evaluate. In ***Temporal-NLI*** (Thukral et al., 2021) dataset, each sample consists of a premise (e.g., "They got married on Saturday") and a hypothesis (e.g., "They got married before Friday"), requiring the model to determine if the relationship is entailment, contradiction, or neutral. ***SpaRTUN*** (Mirzaee & Kordjamshidi, 2022) dataset is designed for spatial reasoning, and it includes stories describing the spatial positions of objects, followed by questions about the orientation of one object relative to another (e.g., left, right, inside, above).

|  | GPT-4o | GPT-4 | Jamba | Mamba | S4-H3 |
|---|---|---|---|---|---|
| Math-QA | 51.8% | 51.0% | 42.2% | 35.0% | 28.6% |
| BIG-Bench Hard | 56.8% | 58.4% | 78.2% | 67.0% | 60.6% |
| Temporal-NLI | 79.4% | 77.2% | 69.8% | 59.2% | 54.6% |
| SpaRTUN | 80.8% | 61.4% | 50.8% | 42.2% | 35.2% |

Table 1: Performance of Attention, SSM and Attention-SSM based models on various function composition tasks involving logical expressions, temporal reasoning, spatial reasoning, and math tasks.

|  | GPT-4o | GPT-4 | Jamba | Mamba | S4-H3 |
|---|---|---|---|---|---|
| Algebra | 51% | 47% | 42% | 36% | 29% |
| Calculus | 50% | 48% | 41% | 34% | 28% |
| Combinatorics | 88% | 70% | 48% | 38% | 33% |
| Game theory | 30% | 40% | 50% | 41% | 32% |
| Trigonometry | 40% | 50% | 30% | 26% | 21% |

Table 2: Performance of models on various topics within the Math-QA dataset. Input dependency consistently improves performance, with Mamba consistently outperforming S4-H3.

The results presented in Tables 1 and 2 highlight several critical observations regarding the performance of various models across different composition tasks. Notably, Mamba (Gu & Dao, 2023) consistently outperforms the S4-H3 (Gu et al., 2022; Fu et al., 2023) model, despite both having almost the same number of parameters. This performance gap underscores the importance of input-dependence in model design, as Mamba's architecture better leverages input information to achieve superior results. Additionally, while GPT-4o is the most performant overall, it struggles with many tasks, including those that seem simple to humans, such as logical expression chaining, as evidenced by its performance on the BIG-Bench Hard (Suzgun et al., 2022) benchmark. This indicates that even state-of-the-art models like GPT-4o have limitations in solving complex composition tasks, which numerically justifies our theoretical findings. Accuracy for all models is calculated as the number of correct answers divided by the total number of samples.

**Compositional tasks** Having demonstrated that models encounter difficulties even with more straightforward composition tasks, we now examine their performance on more complex compositional tasks. Given their proven inability to perform function composition, as established in Theorem 1, it is entirely anticipated that their performance on these tasks will be suboptimal. We explore three compositional tasks: (i) multi-digit multiplication, (ii) dynamic programming, and (iii) Einstein's puzzle.

For the *multi-digit multiplication* task, we generate question-answer pairs such as "What is 5 times 90?" with the answer being "450". This task involves multiplying two numbers, $x = (x_1, x_2, \ldots, x_k)$ and $y = (y_1, y_2, \ldots, y_k)$, where each number can have up to $k$ digits. Consequently, there are $9 \times 10^{(k-1)}$ possible combinations for each number. In our experiments, we set $k$ to 5 and found that both Attention and SSM-based models are unable to solve the 5-by-5 digit multiplication task, even in the case of GPT-4o with CoT prompting (A- 12).

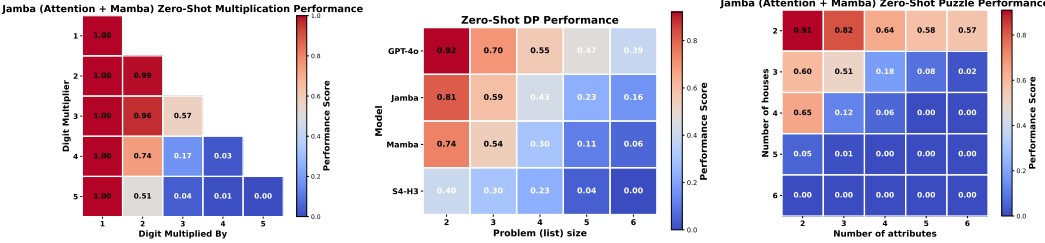

Figure 2: Jamba Lieber et al. (2024) performance on multiplication, DP and puzzle tasks. For DP various models are shown. All struggle with compositional tasks, especially for larger input size.

*Dynamic programming (DP)* recursively decomposes complex problems into simpler sub-problems, making solutions compositional by nature. We consider a classic relaxation of the NP-complete Maximum Weighted Independent Set problem (Kleinberg & Tardos, 2005): *Given a sequence of integers, find a subsequence with the highest sum such that no two numbers in the subsequence are adjacent in the original sequence.* DP can solve This relaxation in $O(n)$ time. For our experiments, we restrict each integer to the range $[-5, 5]$ and follow the generation steps as in (Dziri et al., 2023), with an input list containing from 2 to 6 elements. Prompting details are shown in the A-C.3.

*Einstein's puzzle* is a well-known logic puzzle commonly used as a benchmark for solving constraint satisfaction problems (Prosser, 1993). It involves a series of houses with various attributes, and the objective is to determine which attributes correspond to each home by interpreting a set of predefined natural language clues or constraints. The solution to the puzzle is represented as a matrix of size $H \times A$, where $H$ denotes the number of houses and $A$ represents the number of attributes. As $H$ and $A$ increase, synthesizing partial solutions that satisfy individual constraints becomes increasingly compositionally complex. Qualitative examples and details about data generation for this task are provided in the A-C.2.

## B.2 CoT EXPERIMENTS

Next, we evaluate how the popular chain-of-thought (CoT) prompting method (Wei et al., 2022) affects the performance of GPT-4o (OpenAI, 2023), Jamba (Lieber et al., 2024), Mamba (Gu & Dao, 2023) and S4-H3 (Gu et al., 2022) models on compositional tasks from Sec. B.1. CoT improves performance but does not solve the problem. Details of the experiments and examples of full prompts can be found in the A-D.

## B.3 ERROR ANALYSIS

We focus on graph analysis of errors, emphasizing multi-digit multiplication because this problem is easier to interpret and understand. From this analysis, we obtain a few interesting conclusions about how errors happen and then propagate inside SSM-based LLMs (Fu et al., 2023; Gu & Dao, 2023; Mirzaee & Kordjamshidi, 2022).

**Computation Graph**     To study the propagation of errors and its dependency on input size, we define $A$ as a deterministic algorithm (function) and $\mathcal{F}_A$ as the set of primitives (functions) the algorithm employs during execution. Given inputs $\mathbf{x}$ to the algorithm $A$, we define the static computation graph of $A(\mathbf{x})$, denoted as $G_{A(\mathbf{x})}$, as $G_{A(\mathbf{x})} = (V, E, s, op)$, a directed acyclic graph.

Nodes $V$ represent all variable values during $A$'s execution, where each node $v \in V$ has an associated value $s(v) \in \mathbb{R}$. Edges $E$ represent function arguments involved in computations: for each non-source node $v \in V$, let $U = \{u_1, \ldots, u_j\} \subset V$ be its parent nodes. Then, $s(v) = f(u_1, \ldots, u_j)$ for some $f \in \mathcal{F}_A$. Since each node $v$ is uniquely determined by the computation of a single primitive $f$, we define $op : V \to \mathcal{F}_A, op(v) = f$ as the operator function that yields $s(v)$. Let $S \subset V$ be the source nodes of $G_{A(\mathbf{x})}$, and without loss of generality, let $o \in V$ be its sole leaf node. By definition, $S \equiv \mathbf{x}$ and $A(\mathbf{x}) = s(o)$, representing the input and output of $A$, respectively. To evaluate a language model's ability to follow algorithm $A$, we must linearize $G_{A(\mathbf{x})}$ (arrange the nodes in a linear sequence that respects the dependencies). This means if a node $u$ is a parent of node $v$, the $u$ should appear before $v$ in the sequence. Since we only consider autoregressive models, this linearization must also be a topological ordering. A topological order ensures that every node appears after its parent nodes, maintaining the correct order of computations. This is crucial for correctly following the sequence of operations defined by the algorithm $A$.

To instantiate $G_{A(\mathbf{x})}$, let $\mathcal{F}_A = \{$one-digit multiplication, sum, mod 10, carry over, concatenation$\}$. Source nodes $S$ are digits of input numbers, leaf node $o$ is the final output, and intermediate nodes $v$ are partial results generated during the execution of the long-form multiplication algorithm (see Fig. 3). The corresponding algorithm is on the left of the Fig. 3 - Alg. 1.

**Error propagation**     We examine errors in SSMs, focusing on how they propagate through computation steps. Fig. 4 shows an example from the S4-H3 model performing multi-digit multiplication using CoT prompting. In this case, the model multiplies 9 by 63. It correctly computes $9 \times 3 = 27$ but mistakenly carries over '3' instead of '2', leading to an incorrect middle digit in the final answer

**Algorithm 1** Multiply two numbers

```
 1: function MULTIPLY(x[1..p], y[1..q])          ▷
    multiply x for each y[i]
 2:     for i = q to 1 do
 3:         carry = 0
 4:         for j = p to 1 do
 5:             t = x[j] × y[i]
 6:             t += carry                ▷ add carry
 7:             carry = t ÷ 10
 8:             digits [j] = t  mod 10
 9:         end for
10:         summands[i] = digits
11:     end for
12:     product = Σ_{i=1}^{q} summands[q + 1 − i] ·
        10^{i−1}
13:     return product
14: end function
```

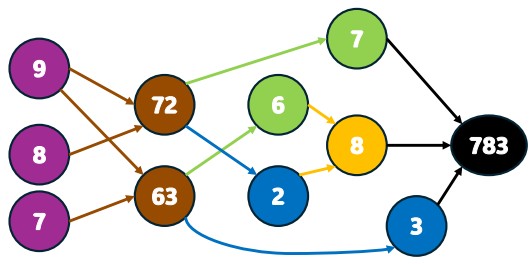

Figure 3: Example of 2-by-1 digit multiplication ($87 \times 9$). Operations on graph include: inputs, multiply 1-digit, carry, sum, mod 10 and output.

despite correct addition in later steps. This highlights *propagation errors*, where an initial mistake affects later steps. Our analysis shows these errors are 2-4 times more common than local errors, consistent with findings from Dziri et al. (2023). This suggests SSMs handle single-step tasks well but struggle with multi-step reasoning, leading to compounded errors.

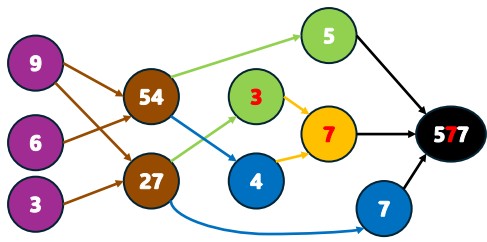

Figure 4: Error Propagation. Carry operation outputs number 3 instead of 2 from node '27', and that error is further propagated, yielding incorrect solution in the middle digit, although all other steps were done right.

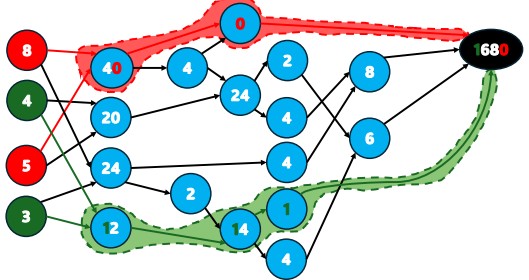

Figure 5: SSMs and Transformers learn shortcuts that seem to solve function composition but fail with larger inputs and out-of-distribution data.

**SSMs learn shortcuts**   The performance of SSMs provides valuable insights into their behavior. These models often predict partially correct answers even when the overall response is incorrect. For example, using Mamba (Gu & Dao, 2023) for 2-by-2 digit multiplication, the first and last digits are usually accurate. The first two and last two digits in larger multiplications tend to be correct. Using Relative Information Gain (RIG) analysis (Dziri et al., 2023), we find that SSMs learn shortcuts, performing fewer operations (illustrated by the red and green subgraphs in Fig. 5). This allows them to frequently predict peripheral digits correctly. For instance, the model multiplies 8 and 5 to compute the last digit, carrying 0 to the end, mimicking human multiplication, and accurately predicting the last digit. RIG analysis reveals a strong correlation between the first digit (or first two digits) of the output and the first digit (or first two digits) of the input numbers.

These models leverage task distribution shortcuts to guess partial answers without whole multi-step reasoning. Increasing the number of Chain-of-Thought (CoT) steps doesn't constantly improve results, especially for larger input sizes (deeper computation graphs). If the model encounters relevant subgraphs during training, its inference seems highly compositional but is based on shortcuts (Geirhos et al., 2020; Liu et al., 2023a; Tang et al., 2023; Du et al., 2022). These experiments indicate that when an output element heavily relies on a few input features, SSMs recognize this correlation during training and map these features to predict the output during testing. This gives the false impression of performing compositional reasoning while bypassing rigorous multi-hop reasoning (Yang et al., 2024).

```
To multiply two numbers, start by multiplying the rightmost digit of
the multiplicand by each digit of the multiplier, writing down the
products and carrying over any remainders. Repeat this process for each
digit of the multiplicand, and then add up all the partial products to
obtain the final result. Here are examples:

Question: what's 32 times 8? Answer 256.
Question: what's 69 times 3? Answer 207.
Question: what's 93 times 6? Answer 558.

Question: what's 76 times 8? Answer:
```

Figure 6: Example prompt for the multiplication task used for the few-shot prompting.

### B.4 LEARNING ALGORITHMIC COMPOSITIONS

Finally, we conduct a comprehensive analysis of the capabilities of SSM-based models, along with GPT-4o (OpenAI, 2023), to "learn" discrete algorithms. This analysis is performed using two tasks that require the composition of multiple discrete sub-tasks. By empirically examining the models' algorithmic learning through compositionality testing, we observe their inability to effectively perform these tasks, even when provided with few-shot prompts and CoT examples (Wei et al., 2022). This suggests that within the framework of in-context learning, SSM and Transformer-based models fail to attain compositional learning when constrained to a fixed number of samples. Details in the A-E.

## C COMPOSITIONAL TASKS DETAILS

### C.1 MULTIPLICATION

We show examples of few-shot and CoT prompting methods for multiplication task (Figs. 6 & 7).

### C.2 EINSTEIN'S PUZZLE

**Data Construction**    Following Dziri et al. (2023), we first define a set of properties such as "Color", "PhoneModel", and "Pet", along with their corresponding values in natural language templates (e.g., "The house has a red color."). We then create a basic and clear set of clue types:
1. **found_at**: For example, "Alice lives in House 2."
2. **same_house**: For example, "The person who is a cat lover lives in the house that has a red color."
3. **direct_left**: For example, "The person who has a dog as a pet lives to the left of the person who lives in a red house."
4. **besides**: For example, "The person who has a dog as a pet and the person who has a red house live next to each other."

Additionally, we introduce more challenging clue types for auxiliary experiments, such as `not_at`, `left_of` (not necessarily directly left), and `two_house_between`. These harder clues are used to test the robustness and versatility of our models.

**Graph Construction**    To address the complex compositional reasoning required for a logical grid puzzle, we utilize existing puzzle solvers de Moura & Bjørner (2008) to generate the computation graph. Our algorithm follows a basic greedy principle: it applies the minimum number of rules necessary to solve any cell. Specifically, if a single rule can solve a cell, that rule is applied.

The algorithm iterates through all clues in the clue set, seeking combinations that can solve any cell in the table. Although this approach may not be the most efficient, it enables models to have explicit scratchpad verbalization via an intuitive computation graph. Fig 8 shows an example of a scratchpad.

**Question:** What is 904 times 74?

**Scratchpad:** Let's perform the multiplication step by step:

Let's multiply 904 by the digit in the ones place of 74, which is 4.

1. Multiply 4 by the digit in the ones place of 904, which is 4. This gives 4 x 4 = 16. Write down the result 6 and carry over the 1 to the next step.
2. Multiply 4 by the digit in the tens place of 904, which is 0. Add the carryover from the previous step to account for this. This gives (0 x 4) + 1 = 1. Write down the result 1.
3. Multiply 4 by the digit in the hundreds place of 904, which is 9. This gives 9 x 4 = 36. Write down the result 36.
4. The partial product for this step is A=3616 which is the concatenation of the digits we found in each step.

Now, let's multiply 904 by the digit in the tens place of 74, which is 7.

5. Multiply 7 by the digit in the ones place of 904, which is 4. This gives 4 x 7 = 28. Write down the result 8 and carry over the 2 to the next step.
6. Multiply 7 by the digit in the tens place of 904, which is 0. Add the carryover from the previous step to account for this. This gives (0 x 7) + 2 = 2. Write down the result 2.
7. Multiply 7 by the digit in the hundreds place of 904, which is 9. This gives 9 x 7 = 63. Write down the result 63.
8. The partial product for this step is B=6328 which is the concatenation of the digits we found in each step.

Now, let's sum the 2 partial products A and B, and take into account the position of each digit: A=3616 (from multiplication by 4) and B=6328 (from multiplication by 7 but shifted one place to the left, so it becomes 63280). The final answer is 3616 x 1 + 6328 x 10 = 3616 + 63280 = 66896.

Figure 7: A sample scratchpad for the multiplication task.

```
This is a logic puzzle. There are 3 houses (numbered 1 on the left, 3
on the right). Each has a different person in them. They have different
characteristics:
- Each person has a unique name: peter, eric, arnold
- People have different favorite sports: soccer, tennis, basketball
- People own different car models: tesla model 3, ford f150, toyota
camry

1. The person who owns a Ford F-150 is the person who loves tennis.
2. Arnold is in the third house.
3. The person who owns a Toyota Camry is directly left of the person
who owns a Ford F-150.
4. Eric is the person who owns a Toyota Camry.
5. The person who loves basketball is Eric.
6. The person who loves tennis and the person who loves soccer are next
to each other.

Let's think step by step. Please first briefly talk about your
reasoning and show your final solution by filling the blanks in the
below table.

$ House: ___ $ Name: ___ $ Sports: ___ $ Car: ___
$ House: ___ $ Name: ___ $ Sports: ___ $ Car: ___
$ House: ___ $ Name: ___ $ Sports: ___ $ Car: ___

Reasoning:
Step 1: First apply clue <Arnold is in the third house.> We know that
The Name in house 3 is arnold.
Step 2: Then combine clues: <The person who loves tennis and the person
who loves soccer are next to each other.> <The person who loves
basketball is Eric.>  Unique Values Rules and the fixed table
structure. We know that The Name in house 1 is eric. The FavoriteSport
in house 1 is basketball. The Name in house 2 is peter.
Step 3: Then apply clue <Eric is the person who owns a Toyota Camry.>
We know that The CarModel in house 1 is toyota camry.
Step 4: Then apply clue <The person who owns a Toyota Camry is directly
left of the person who owns a Ford F-150.> and Unique Values We know
that The CarModel in house 2 is ford f150. The CarModel in house 3 is
tesla model 3.
Step 5: Then apply clue <The person who owns a Ford F-150 is the person
who loves tennis.> and Unique Values We know that The FavoriteSport in
house 2 is tennis. The FavoriteSport in house 3 is soccer.
The puzzle is solved.

Final solution:
$ House: 1 $ Name: Eric   $ Sports: Basketball $ Car: Camry
$ House: 2 $ Name: Peter  $ Sports: Tennis     $ Car: Ford
$ House: 3 $ Name: Arnold $ Sports: Soccer     $ Car: Tesla
```

Figure 8: A sample scratchpad for the Einstein's puzzle task.

## C.3   DYNAMIC PROGRAMMING

We show examples of zero/few-shot and CoT prompting methods for dynamic programming task
(Figs. 9 & 10), following Dziri et al. (2023).

```
Given a sequence of integers, find a subsequence with the highest sum,
such that no two numbers in the subsequence are adjacent in the
original sequence.

Output a list with "1" for chosen numbers and "2" for unchosen ones. If
multiple solutions exist, select the lexicographically smallest. input
= [3, 2, 1, 5, 2].
```

Figure 9: Example prompt for the DP task, used for zero-shot and few-shot settings.

```
Question: Let's solve input = [3, 2, 1, 5, 2].

Scratchpad: dp[4] = max(input[4], 0) = max(2, 0) = 2
dp[3] = max(input[3], input[4], 0) = max(5, 2, 0) = 5
dp[2] = max(dp[3], input[2] + dp[4], 0) = max(5, 1 + 2, 0) = 5
dp[1] = max(dp[2], input[1] + dp[3], 0) = max(5, 2 + 5, 0) = 7
dp[0] = max(dp[1], input[0] + dp[2], 0) = max(7, 3 + 5, 0) = 8

Finally, we reconstruct the lexicographically smallest subsequence that
fulfills the task objective by selecting numbers as follows. We store
the result on a list named "output".

Let can_use_next_item = True.
Since dp[0] == input[0] + dp[2] (8 == 3 + 5) and can_use_next_item ==
True, we store output[0] = 1. We update can_use_next_item = False.
Since dp[1] != input[1] + dp[3] (7 != 2 + 5) or can_use_next_item ==
False, we store output[1] = 2. We update can_use_next_item = True.
Since dp[2] != input[2] + dp[4] (5 != 1 + 2) or can_use_next_item ==
False, we store output[2] = 2. We update can_use_next_item = True.
Since dp[3] == input[3] (5 == 5) and can_use_next_item == True, we
store output[3] = 1. We update can_use_next_item = False.
Since dp[4] != input[4] (2 != 2) or can_use_next_item == False, we
store output[4] = 2.

Reconstructing all together, output=[1, 2, 2, 1, 2].
```

Figure 10: A sample scratchpad for the DP task.

# D  DETAILS OF CoT EXPERIMENTS

## D.1  MAIN CoT EXPERIMENTS

We plot the performance of Jamba Lieber et al. (2024) on multiplication and puzzle tasks and various models on DP tasks after using CoT.

The leftmost heatmap on Fig. 11 represents the Jamba Lieber et al. (2024) model's multiplication performance, showing a consistently high performance for multipliers of 1 and 2, but a noticeable decline as the multipliers increase, particularly beyond 3. The middle heatmap compares the performance of four models—GPT-4o OpenAI (2023), Jamba Lieber et al. (2024), Mamba Gu & Dao (2023), and S4-H3 Gu et al. (2022); Fu et al. (2023)—on dynamic programming tasks with CoT prompting Wei et al. (2022). GPT-4o OpenAI (2023) consistently outperforms the other models, maintaining high performance even for larger problem list sizes, while the performance of the other models decreases more rapidly. The rightmost heatmap displays Jamba's puzzle-solving performance, indicating high accuracy for simpler puzzles with fewer attributes but a steep decline as the complexity increases. These visualizations highlight that while CoT prompting Wei et al. (2022)

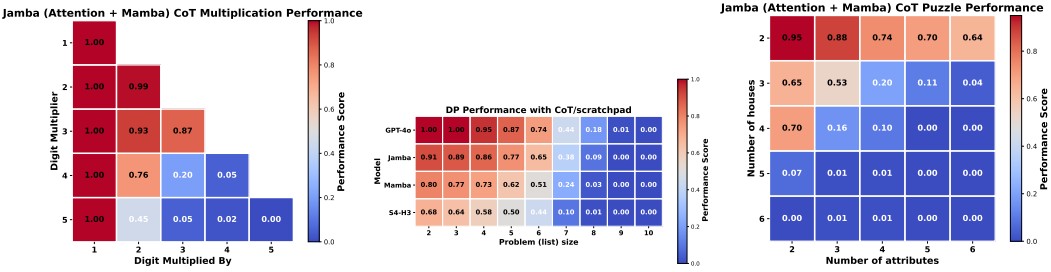

Figure 11: Jamba's Lieber et al. (2024) performance on multiplication and puzzle tasks improves with CoT, though not fully solved. Other models were tested on the DP task, where they failed at higher input sizes, despite CoT.

generally enhances model performance; its effectiveness varies significantly across different models and task complexities.

## D.2 PERFORMANCE OF OTHER MODELS ON MULTIPLICATION AND PUZZLE TASKS

We observe the same pattern on both tasks, for all the models - Figs. 12 & 13. GPT-4o OpenAI (2023) is always the best model, followed by Jamba Lieber et al. (2024), then Mamba Gu & Dao (2023), then S4-H3 Fu et al. (2023); Gu et al. (2022). While CoT helps, it is not enough to solve the task.

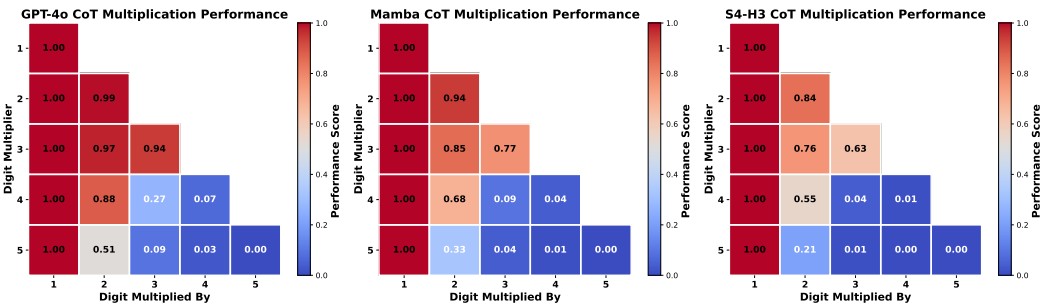

Figure 12: Comparison of different models on multiplication task using CoT.

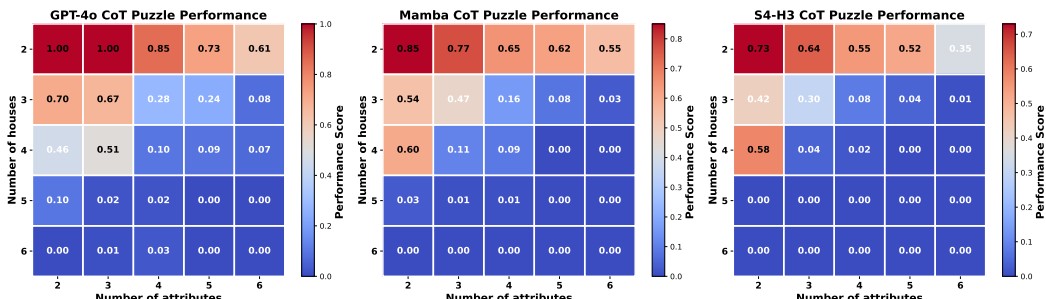

Figure 13: Comparison of different models on puzzle task using CoT.

## D.3 FEW-SHOT PROMPTING MULTIPLICATION RESULTS

We investigate whether few-show prompting (giving a model few input/output pairs) and then asking for the answer to the new problem help. Fig. 14 shows the results, and consistently CoT outperforms Few-shot prompting, and Few-shot prompting outperforms Zero-shot prompting.

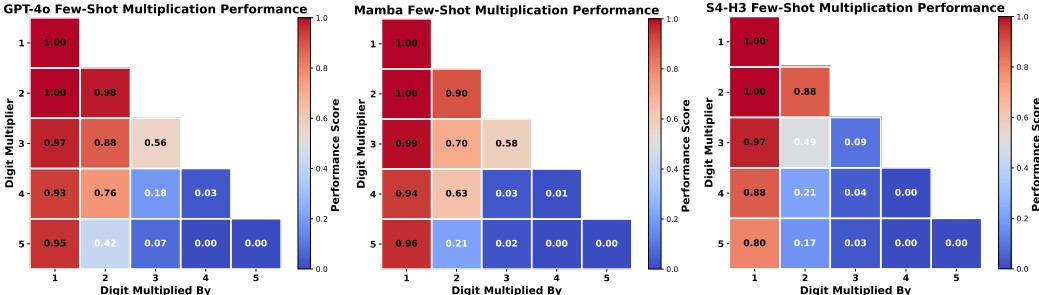

Figure 14: Comparison of different models on multiplication task using few-shot prompting.

# E    ALGORITHMIC COMPOSITIONS

Following Thomm et al. (2024), we evaluate the models using the **PE's Neighbour (PEN)** task. This task involves navigating from one word to the next based on a specified matching criterion and outputting all the neighbors encountered along the way. The PEN task, inspired by Abnar et al. (2023) and rooted in the Pointer Value Retrieval framework Zhang et al. (2022), is particularly compelling due to its four sub-tasks, which test the necessary sub-operations for PEN. These sub-tasks are: (i) Copy words, (ii) Reverse Copy (copying words in reverse order, where words consist of multiple tokens), (iii) PE (outputting words in the matching chain instead of neighbors), and (iv) PE Verbose (PEV) - outputting both the words of the matching sequence and their neighbors. These sub-tasks are essential because, to predict the next word, the model must: take the current word in the answer, obtain the left neighbor (learned in Reverse Copy), match it (learned in PE), and then obtain the right neighbor (learned in Copy). PEV is considered a sub-task because it requires solving the same problem as PEN but with the added complexity of providing both matching words and their neighbors. PEN, on the other hand, only requires outputting the neighbors. For accurate next-token prediction the model cannot simply replicate the last matching sequence word from the previous answers, it must first infer it from the neighbour. To increase the task's complexity, "attention traps" or "doppelgangers" are introduced. These traps create additional matching possibilities by allowing each neighbor to match two other words, thus tempting the model to match from the neighbor of a matching sequence instead. This added layer of difficulty further challenges the models' ability to learn and compose discrete algorithms effectively. **Pointer Execution Reverse Multicount (PER Multi)** shares conceptual similarities with the PEN task; however, instead of matching forward and predicting the current word or its neighbor, the task involves first outputting the last word in the matching sequence and then proceeding backward. Consequently, to accurately predict the first word, the model must identify the end of the matching sequence and output that word. The model needs to count the total number of matchings and the number of matchings that align to the left in the given word order. The answer requires multiplying these two counts, introducing a non-linearity. For this task, we omit any attention traps, as there are no neighbors involved. In the A- E we show concrete prompt examples and share the code.

We conducted extensive evaluations on 500 test samples using various models under different conditions: zero-shot, few-shot (providing a limited number of input-output pairs), and CoT prompting Wei et al. (2022). Remarkably, none of the models, including the state-of-the-art GPT-4o OpenAI (2023), succeeded in solving the PEN task Thomm et al. (2024). Typically, models correctly generated the initial strings but then halted prematurely or produced random strings. The same pattern of failure was observed with the PER Multi-task. Specifically, GPT-4o achieved only 1% and 9% accuracy using few-shot and CoT prompting, respectively, failing to solve the task. The marginal success of GPT-4o is attributed to its substantially larger parameter count compared to SSM-based models (B.1).

Table 3: Model Accuracy for PEN task

| Model | Prompt Setting | Accuracy [%] |
|-------|----------------|--------------|
| GPT-4o | Zero-shot | 0.00 |
|  | Few-shot | 0.00 |
|  | CoT | 0.00 |
| Jamba | Zero-shot | 0.00 |
|  | Few-shot | 0.00 |
|  | CoT | 0.00 |
| Mamba | Zero-shot | 0.00 |
|  | Few-shot | 0.00 |
|  | CoT | 0.00 |
| S4-H3 | Zero-shot | 0.00 |
|  | Few-shot | 0.00 |
|  | CoT | 0.00 |

Table 4: Model Accuracy for PER Multi task

| Model | Prompt Setting | Accuracy [%] |
|-------|----------------|--------------|
| GPT-4o | Zero-shot | 0.00 |
|  | Few-shot | 0.01 |
|  | CoT | 0.09 |
| Jamba | Zero-shot | 0.00 |
|  | Few-shot | 0.00 |
|  | CoT | 0.00 |
| Mamba | Zero-shot | 0.00 |
|  | Few-shot | 0.00 |
|  | CoT | 0.00 |
| S4-H3 | Zero-shot | 0.00 |
|  | Few-shot | 0.00 |
|  | CoT | 0.00 |

In the following subsections, we focus on showing the prompts in few-shot and CoT settings for PEN and PER Multitasks. Moreover, we show the code we used to generate the samples.

### E.1 PROMPTS FOR SSM AND ATTENTION-BASED MODELS FOR PEN TASK

```
Example: eg jy vm3zc si2zf nn4ll zf5ka ki7xd ew0si xp3og il5js xn6yx
my7ec xu2gb if2my fy3so ec2il ob5ch kt5if zc4xp ka3mj og1ud zf2ka yh3ux
hx2kt vc2pf jy4qd lj1xu wy5hx bd4xa my4ec at1kb jy3qd ux1fl ew3si ds2qz
qd7ew xa1ay si1zf ch4lj js3rf fl6xn mj7wy zy6rq zh2gu bj3rb if0my pg5ds
yv3hs zu3ob ta7qi ji2bj mj1wy rq7ul mn3fw ay4qu kt2if kr3qb pr0ah tg0at
uc1vx xd1pd wy4hx dr6fy mk0vj sm0pg jl2mo rb1bd il2js vn6kr km4aq eg7nn
ka6mj qu4vc hx7kt ll2lb ec6il ud2vn di3xs pd6ji qd6ew yx7zu rh4qn lb1ki
js5rf iv3yh jj0fa kb3sm lh6yk so0iv bx6rs qz1vm mw7bm gb2xo uy0ms qb2zy
zm0pz xo4tg zx5jm

Answer: jy ka6mj zf5ka ec6il js5rf ew0si wy4hx qd6ew mj1wy if0my il2js
my4ec si1zf kt2if hx7kt jy4qd

<FEW MORE EXAMPLES>

Your question: ey wt kj5yo jz0aa nu4yw gp2ro mv6kj nk2qz tr3mp ro7rk
tu5xj rk0sj ad2lx up3vd ta7rv qz6ob rc7nt aa4nk mb6mm ob7us jw5wb wt4jz
nn4sr wt0jz ev0fa gp1ro sr1nu sj0ku xs0ta us5up mp6jw vd1gp xj3cs sj7ku
ol3vv vd3gp wd2mv wr4cz dg0py ro5rk jt6ev bv0cf yb2qv ch2ss xa3be nb5id
lx4jt dz5ht wb5wd fb3ax fa0tu jn5ps rv7qj qa7el rn7ad lz3fk mm1tr yd3lv
nt0xs lh4zk mr3ou ja5sn gi5ub rk4sj wm7zm jz3aa be4mb kw3bh qj4xa cg0mi
jl2rn kv1wg qt5mr ye3kg yr5ol nk7qz ub1dg ob3us cs7so gw4vk ey4wm qz2ob
qv4jl xz4hc li0yb oy4qu zm2yr up7vd ou7li rx4wc yw7gi aa2nk yo3qt yz5cx
vv6nn us7up

Clearly mark your answer by writing 'Answer: <your answer>' as last
line.
```

Figure 15: Prompt for the PEN task, showcasing few-shot learning examples. Each word's start and end are encoded as distinct tokens, so a model can pattern-match the respective token to do the matching operation.

**Prompt for the PEN task with few-shot CoT examples and a description.**

I give you a sequence of words. Each word has four characters plus a middle, words are separated by spaces. Start with the leftmost word. Output its neighbor. Then, match the last two characters of the current word (i.e., not the neighbor) to the word starting with those two characters. Again, output the neighbor. Do this until your current word (not the neighbor) has no match anymore.

**Example:** xh jz qw4se zs1qh xv4vn me3af vs1nh ok3ks sn6iv qh1va da5gy ks1ew tw7ik em5zs xs5qu ft3me gt3bc em3zs zn5qv ks5ew by7kn me7af je0wt cb0ft pw6hg rk7cb dv2sn ew3rk yg1by va1cq qu7fp qh4va vn5zn ok1ks cc7tw rk0cb bc7qi jz7em qz2cs ew6rk qv6gt ft7me fp1qw sa6ok sd7pn jz3em wi3da cq7sa iv0vl zs7qh vl2kc va5cq fe5wi xl1zh hg0dv cq4sa ja2nb wh5vv ot4sh qe0jx yt6xs vc0qx nb1am rf2zl kn5hq xg5hk mz7yg aq3uw xh7pw sa7ok wt5ot io6hd pn1je lo6vx hq5cc wp6fc cs7fe yw2ka gy3sd nr0ry am3yt pl0rl ik0tn ub5tq sh0ja ee2it nh6qz xz1ma se0rx is7rn kc1xv cb6ft rx2mz wj7qf.
The leftmost word is **xh**. Its right neighbor is **jz**, so the first output word is **jz**.
Now, we need to find a word that starts with **xh**. The word is **xh7pw**. Its right neighbour is **sa7ok**, so the next output word is **sa7ok**.
Now, we need to find a word that starts with **pw**. The word is **pw6hg**. Its right neighbour is **rk7cb**, so the next output word is **rk7cb**.
Now, we need to find a word that starts with **hg**. The word is **hg0dv**. Its right neighbour is **cq4sa**, so the next output word is **cq4sa**.
Now, we need to find a word that starts with **dv**. The word is **dv2sn**. Its right neighbour is **ew3rk**, so the next output word is **ew3rk**.
Now, we need to find a word that starts with **sn**. The word is **sn6iv**. Its right neighbour is **qh1va**, so the next output word is **qh1va**.
Now, we need to find a word that starts with **iv**. The word is **iv0vl**. Its right neighbour is **zs7qh**, so the next output word is **zs7qh**.
Now, we need to find a word that starts with **vl**. The word is **vl2kc**. Its right neighbour is **va5cq**, so the next output word is **va5cq**.
Now, we need to find a word that starts with **kc**. The word is **kc1xv**. Its right neighbour is **cb6ft**, so the next output word is **cb6ft**.
Now, we need to find a word that starts with **xv**. The word is **xv4vn**. Its right neighbour is **me3af**, so the next output word is **me3af**.
Now, we need to find a word that starts with **vn**. The word is **vn5zn**. Its right neighbour is **ok1ks**, so the next output word is **ok1ks**.
Now, we need to find a word that starts with **zn**. The word is **zn5qv**. Its right neighbour is **ks5ew**, so the next output word is **ks5ew**.
Now, we need to find a word that starts with **qv**. The word is **qv6gt**. Its right neighbour is **ft7me**, so the next output word is **ft7me**.
Now, we need to find a word that starts with **gt**. The word is **gt3bc**. Its right neighbour is **em3zs**, so the next output word is **em3zs**.
Now, we need to find a word that starts with **bc**. The word is **bc7qi**. Its right neighbour is **jz7em**, so the next output word is **jz7em**.
There is no word that starts with **qi**, so we are done with the matching.
**Therefore the answer is:** jz sa7ok rk7cb cq4sa ew3rk qh1va zs7qh va5cq cb6ft me3af ok1ks ks5ew ft7me em3zs jz7em.

<FEW MORE EXAMPLES>

**Your question:** ap cb ch5ya gb6lt uu6le vn0pc og0ef md6ki jx0ph md4ki mq5ox vp1rx zp1xj is5am uq5fb te3rz eq3he cb0md he2zp fe2re ef6yp vn5pc ui3yt kb1ji qg2mq am4vp ez3eq lt5fi hw4eg lz2te wn5kd kb2ji le6wk vp3rx yt3lq rx6gb ey4dx ji3fe lq1dq lz0te wk7sl am6vp zi0up ki5kb ek7uu re0vq cs3ez vq5lz dx6se lt3fi xp2km fe3re bz7hw rx2gb yp6qg gb4lt at4cs fi7vn ox1nl fi5vn ph3zi rz4is kd2bz ji1fe nl3kk ki2kb yo6ey te1rz fd5at qb7ia bn2xp cb4md ya2wn gd7sq xj2jg rp6bl ap1bn is4am se5ui re5vq eg4uq cf6fj fb6jx ll4ic sl4ch qs3nf sp5fd qj6bf dq1og rz1is km6yo vq3lz up5sp wc5iv
**Reason step by step. Clearly mark your answer by writing 'Answer: <your answer>' as last line.**

## E.2 PEN GENERATION CODE

```python
import itertools
import numpy as np
letter_chars = list("abcdefghijklmnopqrstuvwxyz")
big_letter_chars = list("ABCDEFGHIJKLMNOPQRSTUVWXYZ")
number_chars = list("0123456789")
class DataConfig:
    def __init__(self, min_len, max_len, min_hops, max_hops, learn_mode, ambiguous, no_green_confusion):
        self.min_len = min_len
        self.max_len = max_len
        self.min_hops = min_hops
        self.max_hops = max_hops
        self.learn_mode = learn_mode
        self.ambiguous = ambiguous
        self.no_green_confusion = no_green_confusion
    def get(self, key, default):
        return getattr(self, key, default)
class PointerExecutionNeighbour:
    def __init__(self, data_cfg):
        self.length_low = data_cfg.min_len
        self.length_high = data_cfg.max_len + 1
        self.hops_low = data_cfg.min_hops
        self.hops_higher = data_cfg.max_hops + 1
        self.all_2tuples = ["".join(t) for t in itertools.product(letter_chars, repeat=2)]
        self.learn_mode = data_cfg.get("learn_mode", "next")
        self.data_choices = list(number_chars[:8])
        self.ambiguous = data_cfg.get("ambiguous", False)
        self.no_green_confusion = data_cfg.get("no_green_confusion", False)
    def generate_double_pointer_execution(self, n_samples):
        lengths = np.arange(self.length_low, self.length_high)
        samples = []
        answers = []
        while len(samples) < n_samples:
            length = np.random.choice(lengths)
            n_matching_hops = np.random.choice(np.arange(self.hops_low, min(self.hops_higher, length // 2)))
            tuple_choices = np.random.choice(self.all_2tuples, length * 7, replace=False)
            # select the positions where the green matching sequence will be
            positions = np.random.choice(np.arange(1, length), size=n_matching_hops, replace=False)
            cnt = 0
            question_words1 = ["" for _ in range(length)]
            question_words2 = ["" for _ in range(length)]
            remaining_positions = np.random.permutation([i for i in range(1, length) if i not in positions])
            question_words1[0] = tuple_choices[cnt]
            answer_learnseq = [question_words1[0]]
            for pos in positions:
                question_words1[pos] = (tuple_choices[cnt] + np.random.choice(self.data_choices) + tuple_choices[cnt + 1])
                answer_learnseq.append(question_words1[pos])
                cnt += 1
            cnt += 1
            cnt_confuse = cnt + length
            positions_next = np.random.permutation(positions)
            question_words2[0] = tuple_choices[cnt]
            answer = [question_words2[0]]
            # select the positions where the doppelgangers of the neighbours will be
            positions_confuse = np.setdiff1d(np.arange(1, length), positions_next)[0 : len(positions_next)]
            np.random.shuffle(positions_confuse)
            for i, pos in enumerate(positions_next):
                two_big_letters = np.random.choice(self.data_choices, size=2, replace=self.ambiguous)
                question_words2[pos] = (tuple_choices[cnt] + two_big_letters[0] + tuple_choices[cnt + 1])
                question_words2[positions_confuse[i]] = (tuple_choices[cnt] + two_big_letters[1] + tuple_choices[cnt + 1])
                answer.append(question_words2[pos])
                cnt += 1
                cnt_confuse += 1
            cnt = max(cnt, cnt_confuse) + 1
            remaining_next_positions = np.random.permutation([i for i in range(1, length) if i not in positions_next and \
            i not in positions_confuse])
            for pos in remaining_positions:
                question_words1[pos] = (tuple_choices[cnt] + np.random.choice(self.data_choices) + tuple_choices[cnt + 1])
                cnt += 1
                if self.no_green_confusion:
                    cnt += 1
            cnt += 1
            for pos in remaining_next_positions:
                question_words2[pos] = (tuple_choices[cnt] + np.random.choice(self.data_choices) + tuple_choices[cnt + 1])
                cnt += 2
            answer_learnnext = [question_words2[0]]
            for pos in positions:
                answer_learnnext.append(question_words2[pos])
            answer_seqnext = []
            for i in range(len(answer_learnseq)):
                answer_seqnext.append(answer_learnseq[i])
                answer_seqnext.append(answer_learnnext[i])
            answer.reverse()
            question_words = []
            for i in range(length):
                question_words.append(question_words1[i])
                question_words.append(question_words2[i])
            question_str = (f"pe {self.learn_mode}: " + " ".join(["".join(x) for x in question_words]) + " answer: ")
            samples.append(question_str)
            if self.learn_mode == "seq":
                answers.append(" ".join(answer_learnseq))
            elif self.learn_mode == "seqnext":
                answers.append(" ".join(answer_seqnext))
            elif self.learn_mode == "next":
                answers.append(" ".join(answer_learnnext))
        return samples, answers
    def generate(self, n_samples):
        samples, answers = self.generate_double_pointer_execution(n_samples)
        return samples, answers
```

Figure 16: Code utilized for generating instances of the PEN task and its associated subtasks. The hyperparameters employed include a length ranging between $[40, 50]$ and a number of hops ranging between $[10, 20]$.

## E.3 PER MULTI-GENERATION CODE

```python
import itertools
import numpy as np
letter_chars = list("abcdefghijklmnopqrstuvwxyz")
class DataConfig:
    def __init__(self, min_len, max_len, logname, learn_mode="seq"):
        self.min_len = min_len
        self.max_len = max_len
        self.logname = logname
        self.learn_mode = learn_mode
    def get(self, key, default):
        return getattr(self, key, default)
class PointerExecutionReverseMulticount:
    def __init__(self, data_cfg):
        self.length_low = data_cfg.min_len
        self.length_higher = data_cfg.max_len + 1
        self.logname = data_cfg.logname
        self.all_2tuples = ["".join(t) for t in itertools.product(letter_chars, repeat=2)]
        self.learn_mode = data_cfg.get("learn_mode", "seq")
        assert self.learn_mode in ["seq", "multiseq", "seqrev", "multiseqrev"]
    def generate_samples(self, n_samples):
        lengths = np.arange(self.length_low, self.length_higher)
        samples = []
        answers = []
        for _ in range(n_samples):
            length = np.random.choice(lengths)
            tuple_choices = np.random.choice(self.all_2tuples, length + 3, replace=False)
            last_word = tuple_choices[-3] + tuple_choices[-2]
            shuffled_tuple_choices1 = np.random.permutation(tuple_choices[:-3])
            shuffled_tuple_choices2 = np.random.permutation(tuple_choices[:-3])
            words = [ch1 + ch2 for ch1, ch2 in zip(shuffled_tuple_choices1, shuffled_tuple_choices2)]
            start = np.random.choice(words)
            words.append(last_word)
            if "rev" not in self.learn_mode:
                answer = self.solve_seqnext(words, start, self.learn_mode)
            else:
                # change the 2tuple of the start of the start word to a random one
                idx = words.index(start)
                words[idx] = tuple_choices[-1] + words[idx][2:]
                start = words[idx]
                answer, answer_n_left = self.solve_seqnext(words, start, self.learn_mode)
                if self.learn_mode == "seqrev":
                    answer = reversed([f"{w}" for i, w in enumerate(answer)])
                if self.learn_mode == "multiseqrev":
                    answer = reversed([f"{w}.{i*n}" for i, (w, n) in enumerate(zip(answer, answer_n_left))])
            question = (f"prand {self.learn_mode}: " + " ".join(words) + " | " + start + " answer: ")
            samples.append(question)
            answers.append(" ".join(answer))
        return samples, answers
    def solve_seqnext(self, words, start, mode):
        answer_next = []
        matching_seq = []
        current_word = start
        idx = words.index(current_word)
        n_left = 0
        answer_n_left = []
        while True:
            matching_seq.append(current_word)
            answer_next.append(words[idx + 1])
            answer_n_left.append(n_left)
            next_word = [(w, i) for i, w in enumerate(words) if w.startswith(current_word[-2:])]
            if len(next_word) == 0 and "rev" in mode:
                break
            assert len(next_word) == 1
            current_word, new_idx = next_word[0]
            if new_idx < idx:
                n_left += 1
            idx = new_idx
            if current_word in matching_seq:
                break
        if "rev" in mode:
            return matching_seq, answer_n_left
        if "multi" in mode:
            answer = []
            for i, (w, n) in enumerate(zip(matching_seq, answer_n_left)):
                answer.append(f"{w}.{i*n}")
            return answer
        return matching_seq
    def generate(self, n_samples):
        samples, answers = self.generate_samples(n_samples)
        return samples, answers
```

Figure 17: Code employed for generating instances of the Pointer Execution Reverse Multicount task and its associated subtasks. The hyperparameters employed include a length ranging between [10, 20].

