# OpenReview forum: "Limits of Deep Learning: Sequence Modeling through the Lens of Complexity Theory"
_ICLR.cc/2025/Conference — ICLR 2025 Poster_

### Official Review · Reviewer_U9LY · 2024-11-01

**Soundness:** 3
**Presentation:** 3
**Contribution:** 3
**Rating:** 8
**Confidence:** 3

**Summary:**

The authors investigate the computational limitations of structured state space models to perform function composition. They provide theoretical and empirical evidence for the infeasibility of these architectures to perform function composition over large domains.

**Strengths:**

* Well written
* Well referenced
* Settles important questions about the limitations of current architectures of interest for sequence modeling.

This kind of theoretical work is very much needed.

**Weaknesses:**

* The main manuscript could use more background on communication complexity concepts and techniques, presented intuitively, to make the methodology and results more accessible to a general ML audience.
* Computational complexity work often must deal with a gap between formal and practical problems (even if the work is conducted rigorously). The paper could use a discussion about this possible gap, the (non-)robustness of the results to closing this gap, and what kind of studies are needed to close this gap.

Since there is still roughly 1 page of extra space before the manuscript reaches the page limit, it would be useful to have more background on the computational complexity concepts and techniques deployed in the proofs. This could include concepts and techniques from communication complexity, relevant problem classes and reducibilities, and the rationale that supports the proofs.

Computational complexity work often must deal with objections related to the gap between theory and practice. It would be useful to elaborate on how these gaps might play out in this work, and what responses to possible objections might look like. For instance, what aspects of the formal definitions might be overly general with respect to a possibly more restricted practical setting? Might such a discrepancy between formalization and practical scenario account for the results? What kind of descriptive empirical work could be needed to close the gap between formal and practical problems?

**Questions:**

Minor comments and suggestions:

There is a formatting error causing the heading of section 7 to come too close to the previous paragraph.

A summary figure of the empirical results could also be useful in the main manuscript if space limitations allow.

The “Implications for General Artificial Intelligence” paragraph seems more suited to the Conclusions section or a separate “Implications” section than to the “Related Work” section. An Implications section could also elaborate on the links between the formal proofs and the issues of practical interest.

---

> ### Author Response · Authors · 2024-11-25
> **Response to Reviewer U9LY [Part I]**
>
> We sincerely thank Reviewer U9LY for the detailed and thoughtful feedback on our paper. We are delighted that you found the manuscript well-written, well-referenced, and addressing significant questions about the limitations of current architectures for sequence modeling. Your insights have been highly constructive, and we are grateful for the opportunity to improve/rewrite our work based on your suggestions.
>
> ---
>
> ### Addressing Weaknesses
>
> #### 1. **Providing Additional Background on Communication Complexity**
>
> We appreciate your recommendation to improve the accessibility of our manuscript by providing more background on communication complexity. To address this, we will make the following changes in the revised paper:
> - **Introduction of Key Concepts:** We will include a high-level, intuitive explanation of communication complexity, covering core ideas like communication protocols, the pointer chasing problem, and their relevance to sequence modeling.
> - **Background on Relevant Problem Classes:** The revised paper will provide additional context for computational classes such as $\mathbf{L}$ (logarithmic space) and $\mathbf{NL}$ (nondeterministic logarithmic space), explaining their role in characterizing the challenges faced by sequence models.
>
> These additions will ensure that readers without a computational complexity background can better understand the motivations and implications of our work.
>
> ---
>
> #### 2. **Bridging the Gap Between Theory and Practice in Computational Complexity**
> We value your observations about the discrepancies between theoretical constructs and practical applications. Below, we address your specific questions:
>
> **(a) What aspects of the formal definitions might be overly general with respect to practical settings?**
>
> The following assumptions in our theoretical framework contribute to potential overgeneralization:
>
> - **Worst-Case Analysis:** Theoretical results often reflect worst-case scenarios, which may not represent the typical distributions of real-world data.
> - **Simplified Architectures:** We consider simplified architectures for analytical tractability, omitting practical enhancements such as optimized attention mechanisms or regularization strategies.
> - **Deterministic Computation:** Our theoretical results assume deterministic computations, overlooking stochastic elements like dropout and random initialization.
> - **Perfect Inputs:** Our analysis does not model Real-world data imperfections (e.g., noise, errors).
>
> But, since models are proven to be limited even in these idealized cases, they will struggle even more in practice when there is more stochasticity.
>
> **(b) Could discrepancies between formalization and practice account for the observed results?**
> Yes, these discrepancies might explain some of the divergence between theoretical and practical performance. However, our empirical findings suggest that theoretical limitations remain relevant:
> - **Performance Degradation:** Tasks requiring deep compositional reasoning showed significant performance degradation, aligning with our theoretical predictions.
> - **Error Patterns:** Failure modes observed in practice are consistent with our identified theoretical constraints.
>
> **(c) What empirical work could help bridge this gap?**
> To close the gap between theory and practice, we propose the following empirical strategies:
> - **Controlled Experiments:** Design benchmarks tailored to theoretical limitations, allowing for direct empirical validation.
> - **Ablation Studies:** Investigate how specific architectural components and training strategies impact model performance on theoretical tasks.
> - **Error Analysis:** Conduct detailed examinations of failure cases to uncover reasoning bottlenecks in models.
> - **Cross-Model Comparisons:** Compare various architectures (e.g., RNNs, Transformers, and SSMs) to identify universal versus architecture-specific limitations.
> - **Real-World Task Evaluation:** Assess models on real-world problems requiring compositional reasoning to determine practical applicability.
> - **Benchmark Development:** Establish standardized datasets and tasks to improve broader comparisons and progress tracking.
> - **Theory-Informed Experiments:** Use theoretical insights to design experiments that test specific hypotheses about model limitations.
>
> Although we included most of them, these initiatives would promote a deeper understanding of the interplay between theoretical constraints and practical model performances. They can be interesting follow-up work. Thank you for pointing out this direction.
>
> ---

---

> ### Author Response · Authors · 2024-11-25
> **Response to Reviewer U9LY [Part II]**
>
> ### Addressing Specific Comments and Suggestions
>
> #### Formatting Error in Section 7 Heading
> Thank you for identifying this oversight. We will fix the spacing issue to ensure the heading is properly formatted.
>
> #### Relocating the "Implications for General Artificial Intelligence" Paragraph
> We appreciate your suggestion to improve the flow of this discussion. We will relocate this paragraph to a dedicated "Implications" section or incorporate it into the "Conclusion" to improve the paper's structure.
>
> ---
>
> Thank you once again for your valuable feedback. We are confident these revisions will make our work better and more accessible to the community.

---

> > ### Comment · Reviewer_U9LY · 2024-11-25
> > **Quick reaction to the rebuttal**
> >
> > Thanks for your outline of the proposed changes.  I think they look good and I don't have any more comments at this time.
> >
> > ------
> >
> > Note that you can already make changes to the manuscript and update your submission (this has been open since the beginning of the discussion period). This way reviewers can see the actual changes rather than a proposal for changes, and possibly reassess their evaluation. (This is just my impression and not an official recommendation).

---

### Official Review · Reviewer_Lnbo · 2024-11-03

**Soundness:** 4
**Presentation:** 4
**Contribution:** 4
**Rating:** 8
**Confidence:** 3

**Summary:**

The core result of this paper is a proof that one-layer structured state space models cannot perform efficient function composition. The paper builds upon prior work from from Christos Papadimitriou in Peng et all who proposed the paradigm. The paper extends results to SSM models and shows that significant COT computation is required to achieve function composition in SSMs. Authors extend analysis to multi layer SSMsdemonstrating that the computation of an L-layer SSM on a prompt of length N can be carried out using O(L log N ) bits of memory, positioning SSMs within the complexity class L (logarithmic space). This implies that SSMs cannot solve problems that are NL-complete unless L = NL, which is widely believed to be false.

**Strengths:**

Paper is an important contribution as it extends rigorous complexity theory analysis of LLM model limitations to SSM models proving sharp results on resources required to execute function composition.

**Weaknesses:**

No real weakness. Potentially authors could spend more time comparing their results to Peng for readers.

**Questions:**

Authors could also clarify the numerical experiments and their relationship to the main theorem with respect to layer number.

---

> ### Author Response · Authors · 2024-11-25
> **Response to Reviewer Lnbo**
>
> We sincerely thank the reviewer for their positive feedback and highlighting our paper's contributions. Below, we address your suggestions regarding comparisons to prior work and the relationship between numerical experiments and theoretical results.
>
> ---
>
> ### **Comparing Our Results to Peng et al.**
>
> **Comment:** *No real weakness. Potentially authors could spend more time comparing their results to Peng for readers.*
>
> **Response:**
> We agree that a detailed comparison with Peng et al. would aid readers. Here’s a concise summary of distinctions and extensions provided by our work:
>
> 1. **Architectural Scope:**
>    - Peng et al. focused on Transformers, showing limitations in function composition over large domains.
>    - We extended this analysis to Structured State Space Models (SSMs), proving that they also struggle with function composition without exponential state dimensions, suggesting broader limitations across sequence models.
>
> 2. **Theoretical Contributions:**
>    - Peng et al. placed Transformers in weak complexity classes, showing their inefficiency in compositional reasoning.
>    - We showed that multi-layer SSMs operate within \(L\) (logarithmic space), reinforcing their computational constraints and inability to solve \(NL\)-complete problems unless \(L = NL\).
>
> 3. **Chain-of-Thought (CoT) Analysis:**
>    - Peng et al. highlighted exponential scaling in reasoning steps with CoT prompting for Transformers.
>    - We demonstrated similar limitations for SSMs, with CoT requiring insufficient polynomially growing steps to overcome inherent architectural constraints.
>
> 4. **Empirical Evidence:**
>    - Both works showed limitations in function composition tasks, but our experiments confirmed that SSMs face similar barriers despite their distinct architectural design, along with many more experiments in the appendix.
>
> **Summary:**
> Our findings complement Peng et al.’s work by generalizing their insights to SSMs, reinforcing that these limitations are not specific to Transformers but reflect fundamental constraints in sequence modeling architectures.
>
> ---
>
> ### **Clarifying the Relationship Between Numerical Experiments and Theoretical Results**
>
> **Comment:** *Authors could clarify the numerical experiments and their relationship to the main theorem with respect to layer number.*
>
> **Response:**
> Our numerical experiments were designed to validate the theoretical results, particularly Theorem 3, which places \(L\)-layer SSMs in \(L\) (logarithmic space). Key findings include:
>
> 1. **Layer Depth and Performance:**
>    - Increasing SSM depth improved performance marginally but plateaued quickly, consistent with the theoretical prediction that computational capacity does not scale significantly with depth.
>
> 2. **Function Composition Tasks:**
>    - Deeper SSMs did not overcome limitations in multi-step reasoning or function composition, aligning with our proof that they remain constrained by \(L\).
>
> 3. **Chain-of-Thought Prompting:**
>    - CoT prompting aided SSMs but required polynomial growth in reasoning steps. Additional layers did not alleviate this growth, reinforcing that the architectural constraints persist despite CoT.
>
> **Conclusion:**
> These experiments support our theoretical results, demonstrating that SSM limitations stem from fundamental architectural constraints, not just depth.
>
> ---
>
> We appreciate the reviewer’s thoughtful suggestions and hope these clarifications improve the understanding. We remain open to further questions and would be grateful if these responses are satisfactory enough to warrant an updated score.

---

### Official Review · Reviewer_BvEu · 2024-11-04

**Soundness:** 3
**Presentation:** 3
**Contribution:** 3
**Rating:** 6
**Confidence:** 3

**Summary:**

This paper discusses the limitations of the reasoning abilities of SSMs and Transformers. Theoretically, it presents three theorems: (i) the inability of SSMs to efficiently perform function composition; (ii) Chain-of-thought helps yet with exponential increase in reasoning steps; and (iii) the inability of multi-layer SSMs to solve problems that are NL-Complete unless L = N. Empirically, the authors present experiments in qualitative examples of zero-shot inference, function composition, math and other reasoning tasks.

**Strengths:**

- Originality: This paper precisely defines the problem and proposes new theorems showing new results.
- Clarity and rigor: The theoretical and empirical sections are clearly structured, and concepts are clearly defined.
- Significance: Reasoning is a crucial problem in this domain. The lower bound on CoT steps required for iterated function composition is an interesting result, showing the practical challenges in scaling up these techniques, which is valuable for the research community. I am not able to verify the proofs, though.
- Empirical evaluation: The empirical results corroborate the theoretical claims effectively. The authors tested various composition tasks (function, spatial, temporal, and relational), presenting concrete evidence of performance degradation across different types of tasks.

**Weaknesses:**

Note: Unfortunately I am not a domain expert in theoretical computer science, and my evaluations are based on educated guess.

- Could the authors provide an additional section on limitations (of this work per se) and future works, that practitioner may follow? For example, could the authors discuss more specific architectural modifications based on the existing results?
- In practice, complicated reasoning tasks are often solved with (tree) search (cf., [alphaproof](https://deepmind.google/discover/blog/ai-solves-imo-problems-at-silver-medal-level/), [GPT-f](https://arxiv.org/pdf/2009.03393), [HyperTree Search](https://arxiv.org/pdf/2205.11491)), potentially with self-correction (cf., [Self-Correction](https://arxiv.org/pdf/2405.18634), [SCoRe](https://arxiv.org/pdf/2409.12917)), beyond naive stepwise chain-of0thought augmentation. Can the authors provide further discussions on this?

References:
[1] Polu, Stanislas, and Ilya Sutskever. "Generative language modeling for automated theorem proving." arXiv preprint arXiv:2009.03393 (2020).
[2] Lample, Guillaume, Timothee Lacroix, Marie-Anne Lachaux, Aurelien Rodriguez, Amaury Hayat, Thibaut Lavril, Gabriel Ebner, and Xavier Martinet. "Hypertree proof search for neural theorem proving." Advances in neural information processing systems 35 (2022): 26337-26349.
[3] Wang, Yifei, Yuyang Wu, Zeming Wei, Stefanie Jegelka, and Yisen Wang. "A Theoretical Understanding of Self-Correction through In-context Alignment." arXiv preprint arXiv:2405.18634 (2024).
[4] Kumar, Aviral, Vincent Zhuang, Rishabh Agarwal, Yi Su, John D. Co-Reyes, Avi Singh, Kate Baumli et al. "Training language models to self-correct via reinforcement learning." arXiv preprint arXiv:2409.12917 (2024).

**Questions:**

See the weakness. I am happy to raise my score if the authors can address the concerns proposed by reviewers.

---

> ### Author Response · Authors · 2024-11-25
> **Response to Reviewer BvEu [Part I]**
>
> We sincerely thank the reviewer for their thoughtful feedback and insightful suggestions. We appreciate acknowledging our work’s originality, clarity, and significance in exploring the limitations of reasoning abilities in Structured State Space Models (SSMs) and Transformers. Below, we address the reviewer’s queries and expand on their recommendations.
>
> ---
>
> ### **1. Additional Section on Limitations and Future Work**
>
> **Comment:** *Could the authors provide an additional section on limitations (of this work per se) and future works that practitioners may follow? For example, could the authors discuss more specific architectural modifications based on the existing results?*
>
> #### **Response:**
>
> We acknowledge the importance of delineating the limitations of our work and outlining potential directions for future research. Below, we address these in detail:
>
> #### **Limitations of Our Work**
>
> 1. **Focus on Current Architectures:**
>    Our theoretical and empirical analyses are restricted to existing SSM and Transformer architectures. We did not explore architectural modifications or external enhancements that might expand their reasoning capabilities.
>
> 2. **Idealized Assumptions:**
>    The theoretical results are derived under standard complexity-theoretic and algorithmic assumptions. These abstractions do not fully account for practical factors such as optimization dynamics, hardware constraints, or domain-specific training regimes.
>
> 3. **Scope of Empirical Evaluation:**
>    While we conducted extensive experiments across diverse tasks, our exploration is not exhaustive. Alternative configurations, training paradigms, or specific domain-focused benchmarks could reveal additional insights.
>
> ---
>
> #### **Future Work and Potential Architectural Modifications**
>
> Building on our findings, we identify the following directions for future research:
>
> 1. **Integrating External Memory Mechanisms:**
>    - Incorporating memory-augmented components, such as differentiable neural computers or external attention mechanisms, could enable models to effectively store and retrieve intermediate computations.
>    - This approach could mitigate existing multi-step reasoning and function composition limitations by allowing models to scale reasoning over extended sequences. While this improves general reasoning capabilities, it may remain insufficient for problems demanding deeper algorithmic understanding (e.g., solving unsolved mathematical conjectures).
>
> 2. **Incorporating Symbolic Reasoning Components:**
>    - Hybrid architectures that combine neural networks with symbolic reasoning frameworks (e.g., integrating SAT solvers or formal theorem provers) may improve models’ logical inference capabilities.
>    - Symbolic modules can complement neural systems by performing exact computations, thus addressing the constraints of neural-based architectures in handling structured, logical reasoning.
>
> 3. **Implementing Specialized Training Strategies:**
>    - Designing tailored training paradigms, such as curriculum learning or meta-learning approaches, could enable models to progressively develop reasoning skills.
>    - Auxiliary tasks emphasizing multi-step reasoning or function composition might also facilitate improved generalization to complex tasks.
>
> 4. **Exploring Alternative Computational Frameworks:**
>    - Investigating architectures inspired by computational models with theoretically higher capacities (e.g., hypergraph-based neural networks or quantum-inspired architectures) could unlock new pathways for complex reasoning.
>    - Such frameworks may inherently support reasoning-intensive tasks without requiring impractical resource scaling.
>
> 5. **Advancing Neural Algorithmic Reasoning:**
>    - Neural Algorithmic Reasoning, an emerging paradigm that aims to align neural networks with classical algorithmic processes, presents a promising avenue.
>    - By embedding algorithmic structures into neural systems, these models can execute complex computations over large domains (without requiring impractically large state dimensions or excessive computational resources), enabling capabilities such as iterative function composition, mathematical problem-solving, and logical deduction.
>    - Leveraging this approach can address the architectural and training bottlenecks identified in our study, offering a robust framework for tackling reasoning-centric tasks.
>
> ---

---

> > ### Author Response · Authors · 2024-11-25
> > **Response to Reviewer BvEu [Part II]**
> >
> > **2. Discussing Tree Search and Self-Correction Methods in Relation to Our Work**
> >
> > *Question:* *In practice, complicated reasoning tasks are often solved with (tree) search (cf., AlphaProof, GPT-f, HyperTree Search), potentially with self-correction (cf., Self-Correction, SCoRe), beyond naive stepwise chain-of-thought augmentation. Can the authors provide further discussions on this?*
> >
> > **Answer:**
> >
> > We appreciate the reviewer’s insightful question and the opportunity to discuss how advanced methods like tree search and self-correction relate to our findings.
> >
> > ---
> >
> > **Relation to Our Analysis**
> >
> > 1. **Intrinsic Architectural Capabilities:**
> >    Our analysis primarily focuses on the inherent computational limitations of Structured State Space Models (SSMs) and Transformers when used in isolation, without the aid of external mechanisms or augmentations.
> >
> > 2. **External Mechanisms as Augmentations:**
> >    Methods like tree search and self-correction introduce external reasoning frameworks that extend beyond the models' intrinsic capabilities. While they mitigate some computational limitations, they do so by leveraging additional resources or algorithms rather than addressing the core architectural constraints.
> >
> > ---
> >
> > **Tree Search Methods**
> >
> > 1. **Overview:**
> >    Tree search algorithms, such as those employed in AlphaProof, GPT-f, and HyperTree Search, improve reasoning by systematically exploring multiple solution paths. They allow models to navigate combinatorial spaces and evaluate alternatives effectively.
> >
> > 2. **Impact on Reasoning:**
> >    By integrating systematic exploration, these methods enable models to handle complex, structured reasoning tasks requiring deep logical deductions or exploration of solution spaces.
> >
> > 3. **Connection to Our Work:**
> >    - Tree search compensates for the lack of native reasoning capacity in SSMs and Transformers by layering external logic and decision-making.
> >    - Our findings suggest that current architectures struggle with tasks requiring multi-step reasoning or function composition without such mechanisms. The need for tree search highlights the models’ reliance on external processes for tasks beyond their intrinsic computational scope.
> >
> > ---
> >
> > **Chain-of-Thought (CoT) Prompting Limitations**
> >
> > 1. **Naive CoT vs. Advanced Methods:**
> >    Our results show that naive stepwise CoT prompting fails to overcome the computational limits of SSMs and Transformers. Advanced approaches like tree search and self-correction provide additional reasoning capabilities but at the cost of relying on external augmentations.
> >
> > 2. **Implications:**
> >    These methods demonstrate that intrinsic architectural changes, rather than external augmentations, are needed to address reasoning limitations directly.
> >
> > ---
> >
> > **Implications for Future Research**
> >
> > 1. **Architectural Integration of Reasoning Mechanisms:**
> >    A key direction is embedding reasoning frameworks such as search algorithms and iterative refinement into the models themselves, enabling these capabilities to become intrinsic rather than external.
> >
> > 2. **Designing Intrinsically Iterative Models:**
> >    New architectures that inherently support iterative reasoning and dynamic exploration could eliminate the reliance on external augmentations.
> >
> > ---
> >
> > We thank the reviewer for prompting this valuable discussion and will explicitly clarify in our manuscript that our analysis does not incorporate external "engines." We welcome any further questions and sincerely hope that, if the reviewer is satisfied with our responses, they will consider revising their score.

---

> > > ### Comment · Reviewer_BvEu · 2024-11-29
> > >
> > > Thanks for the authors' comments! I will carefully walk through the revisions (also feedback from other reviewers here) during the weekend and will update the rating if the concerns are well-addressed.

---

> > > > ### Author Response · Authors · 2024-12-01
> > > > **Response to Reviewer BvEu [01.12.2024]**
> > > >
> > > > Dear Reviewer BvEu,
> > > >
> > > > Thank you for your time and consideration in reviewing our revisions. We appreciate your thoughtful feedback and are glad our responses have addressed your concerns. We are also grateful that you have updated your score.

---

### Official Review · Reviewer_yPps · 2024-11-04

**Soundness:** 2
**Presentation:** 3
**Contribution:** 3
**Rating:** 6
**Confidence:** 4

**Summary:**

The paper theoretically and empirically studies the limitations of the computational power of SSMs in terms of effective space complexity and their ability to do function composition.

**Strengths:**

Novel theoretical insight on the recently popular SSMs (especially in regards to their limited ability to do function composition), echoing similar previous work on Transformers. Experiments validate the theory.

**Weaknesses:**

1. The proof of Theorem 1 appears to rely on the specific format of the prompt (see Questions below).
2. I have doubts about the correctness of Theorem 4's proof and I don't think the theorem statement itself properly formalizes the insight that it's trying to convey (see Questions below).
3. It seems to me that the criticism in Question 6 below (i.e., that Theorem 4 really says "SSMs' computational power is bounded by the amount of space used to represent floating points") equally applies to the cited papers Merrill & Sabharwal 2023 and Merrill & Sabharwal 2024. Since Theorem 3 is based on the results of the latter paper, it relies on the problematic assumption that precision $p = O(\log N)$. A consequence of this is that changing the assumption on how $p$ scales with $N$ even by a little bit will completely mess up the theorem statement, e.g., if $p = O(\mathrm{poly}(n))$ then we can no longer say that SSMs can't solve these P-complete problems. (See Question 6 below for a more detailed discussion of this point as applied to Theorem 4.)

**Questions:**

The following questions are about Theorem 1.
1. The proof seems to rely on the specific ordering of $g$ followed by $f$ followed by $x$ in the prompt. Does a similar proof work when these pieces are in different orders? In particular, what about the order that would intuitively be the easiest for the SSM: $x$ followed by $g$ followed by $f$? (The intuition here comes from the fact that a streaming algorithm taking in $(x, g, f)$ in this order wouldn't need to store the entire table of $f$ or $g$.)

The following questions are about Theorem 2.
1. How's the definition of CoT different from just autoregressive decoding?

The following questions are about Theorem 4.
1. Lines 74 & 386: Isn't $\mathsf{L}$ a class of decision problems? Shouldn't one say $\mathsf{FL}$ instead of $\mathsf{L}$ here?
2. How is it possible for $p, d$ to grow with $N$? While an SSM can take an input of variable length $N$, its $p$ and $d$ must be fixed, right? I don't know what it means for the precision and hidden dimension of an SSM to increase as the input sequence becomes longer. SSMs are unlike boolean circuits which require a differently-sized circuit for every possible input size.
3. While the theorem statement assumes $p, d = O(\mathrm{poly}(N))$, the proof assumes $d = O(1), p = O(\log N)$. For example, line 359 says "each element of these matrices can be represented using $O(\log N)$ bits." Line 377 says "numbers are represented with $O(\log N)$ bits of precision." Line 379 says "we only need to keep the current and previous hidden states", which require $O(dp)$ space, hence implicitly assuming $dp = O(\log N)$.
4. How are $A_t, B_t, C_t, D_t$ (which are functions of $x_t$) computed? Some assumption on the space complexity of these computations is missing from the theorem statement.
5. This is a small detail, but in the current formulation where the input sequence is a bunch of vectors of real numbers, I believe the input size is actually $Ndp$. But $O(\log N)$ implies $O(\log(Ndp))$, so there's no problem here even when $dp = \omega(1)$ in $N$.
6. Assuming $A_t, B_t, C_t, D_t$ are independent of the input sequence (and "easily computable"), it's easy to generalize the theorem to say that an SSM with $L$ layers, precision $p$ and hidden dimension $d$ is equivalent to an algorithm that uses $O(Lpd)$ space (i.e., you store hidden states $h_t^{(l)}$ of all layers $l$ at the current time step $t$ and the intermediate result $y_t^{(l_\text{cur})}$ of the current layer $l_\text{cur}$). So Theorem 4's statement that SSMs use log-space is actually just an artifact of $Lpd = O(\log N)$ in the assumption of the (corrected version of the) theorem (see Question 3 above). So it's unclear what insight we get from the fact that SSMs are equivalent to log-space when $Lpd = O(\log N)$. If $p, d = O(1)$ (i.e., the case of actual SSMs), then we get that linear SSMs are equivalent to algorithms that use $O(1)$ space, but does that mean that the decision problems that linear SSMs can solve must be regular languages? (A question of a similar nature: my computer has finite memory, so does that mean it can only decide regular languages?) On the other hand, if $p, d = O(\mathrm{poly}(N))$, then we get that linear SSMs can be simulated by algorithms that use polynomial space, and we no longer get the takeaway that SSMs are limited. To summarize, the fact that the computational power of SSMs (as proven using the method in Theorem 4) varies widely based on what is assumed of the scaling of precision $p$ w.r.t. $N$ indicates a failure of "SSM with precision $p = O(f(N))$" as model to accurately capture the behavior of actual SSMs. Theorem 4 generalized would tell me that an SSM with precision $p = 16N$ can potentially solve P-complete problems and yet an SSM with $p = 64$ can only decide regular languages, but in practice there shouldn't be a difference in what these two classes of SSMs can do. (See below for what I think a theorem statement that actually formalizes the intuitive notion of "SSMs can only compute problems in [complexity class]" might look like.)

Here's my suggestion for how to actually formalize "SSMs only have _____ computational power" into a theorem statement to replace the current Theorem 4.

First, we need to define the inputs and outputs as actual strings, since members of $\mathsf{FL}$ are functions $f : \Sigma^* \to \Sigma^*$. So the input to the SSM is a sequence of tokens $w_1 \ldots w_N$ that first get embedded into vectors $x_t \in \mathbb R^m$ ($t \leq N$), and the output is an argmax applied on top of the last layer of $y_t$'s for $t > N$ until the <eos> token. (The $w_t$ for $t > N$ are the decoded $\arg \max_i (y_{t-1})_i$ as usual.) A basic formalization like this to properly define the inputs/outputs of an SSM is missing from the paper and should be added.

Now, to prevent the issue in Question 6 above where the SSM's computational power ends up being limited by the finite precision in floating point computations, we really should assume infinite precision here. (So we're working with an idealized version of an SSM and not a real one.) And then we can ask about the computational power of an SSM with (given) hidden dimension $d$ and # layers $L$. So the theorem statement looks something like "Functions $f : \Sigma^* \to \Sigma^*$ that can be computed by an infinite-precision SSM with hidden dimension $d$ and # of layers $L$ are within the class _____."

The current proof of Theorem 4 doesn't work anymore, since directly carrying out the computations in the SSM would require infinite space. However, intuitively, SSMs should still be limited in their computational power for the following reason. Since input tokens are discrete, intuitively the spaces of $h_t$ and $y_t$ can, in some sense, be discretized into "regions of equivalent behavior". Thus, if we show that under certain conditions, the space of the hidden state can be divided into a finite number of regions of equivalent behavior, then the SSM is just a finite-state machine and the functions it computes are computable in $O(1)$ space.

---

> ### Author Response · Authors · 2024-11-26
> **Response to Reviewer yPps [Part I]**
>
> Dear Reviewer, thank you for your thorough and insightful review of our paper. We appreciate your constructive feedback and have carefully considered each of your points. Before the deadline, we will update the manuscript.
>
> ---
>
> ### **Responses to Questions:**
>
> #### **1. Questions about Theorem 1:**
>
> **Question 1:**
>
> *The proof seems to rely on the specific ordering of g followed by f followed by x in the prompt. Does a similar proof work when these pieces are in different orders? In particular, what about the order that would intuitively be the easiest for the SSM: x followed by g followed by f? (The intuition here comes from the fact that a streaming algorithm taking in \(x, g, f) in this order wouldn't need to store the entire table of f or g.*)
>
> **Response:**
>
> You are correct that the ordering of the prompt in our proof plays a role in the communication complexity argument. However, the fundamental limitation does not depend on the specific order of x, g and f in the prompt.
>
> In our initial proof, we assumed a prompt where the descriptions of g and f precede the query x. This aligns with how we constructed the communication protocol to simulate the SSM's computation.
>
> We can adapt the proof accordingly if the prompt order is x followed by g followed by f. The key challenge remains: to compute f(g(x)) over large domains without storing substantial information about f and g is fundamentally difficult for SSMs due to their limited state size.
>
> Even if x is presented first, the SSM must retain x while processing g and then f. Since g and f are arbitrary functions over large domains, the model must still capture significant information about them to compute f(g(x)). This requires a substantial state size, regardless of the prompt order.
>
> Therefore, the limitations we proved still hold under different prompt orderings. We will revise the proof in our paper to clarify that the argument applies regardless of the order of x, g, and f in the prompt.
>
> **Edits to the Manuscript:**
>
> - **Section 4 (Function Composition Requires Wide One-Layer Models):** We will revise the proof of Theorem 1 to explicitly address different prompt orderings. We will demonstrate how the communication complexity argument can be adapted to various prompt structures, reinforcing that the fundamental limitation arises from the need to represent large functions, not from the specific prompt order.
>
> ---
>
> #### **2. Questions about Theorem 2:**
>
> **Question 2:**
>
> *How is the definition of CoT different from just autoregressive decoding?*
>
> **Response:**
>
> Thank you for highlighting the need for clarification. In our definition, the Chain-of-Thought (CoT) refers to a model generating intermediate reasoning steps that explicitly break down the problem-solving process. While autoregressive decoding involves predicting the next token based on previous tokens, CoT is designed to encourage the model to produce a sequence of logical reasoning steps leading to the final answer.
>
> The key differences are:
>
> - **Purpose:** CoT aims to mimic human-like reasoning by generating intermediate steps that make the model's thought process explicit.
>
> - **Structure:** In CoT, the model is prompted or trained to produce these reasoning steps, which may involve additional tokens that represent sub-calculations or explanations.
>
> - **Autoregressive Decoding:** While CoT uses autoregressive decoding as the mechanism to generate the sequence, not all autoregressive decoding involves CoT. Standard autoregressive models may generate outputs without explicit intermediate reasoning.
>
> In our paper, we defined CoT within the context of SSMs by formalizing how the model recursively generates additional tokens based on its previous outputs and inputs, focusing on the reasoning aspect.
>
> **Edits to the Manuscript:**
>
> - **Section 5 (Many Thought Steps are Needed):** We will update the definition of CoT to clearly distinguish it from standard autoregressive decoding. We will emphasize the role of intermediate reasoning steps in CoT and explain how it simulates multi-step reasoning processes, making the distinction more explicit.
>
> ---
>
> #### **3. Questions about Theorem 4:**
>
> **Question 3:**
>
> *Lines 74 & 386: Isn't L a class of decision problems? Shouldn't one say FL instead of L here?*
>
> **Response:**
>
> You are correct. In computational complexity, L refers to the class of decision problems solvable in logarithmic space, while FL denotes the class of function problems computable in logarithmic space. Since our focus is on function computation rather than decision problems, it is more precise to refer to FL.
>
> **Edits to the Manuscript:**
>
> - **Throughout the Paper:** We will replace L with FL when referring to function computation in logarithmic space. This change will improve the precision of our statements regarding the computational complexity classes involved.
>
> ---

---

> ### Comment · Reviewer_yPps · 2024-11-27
> **Response to authors' rebuttal Part I**
>
> I thank the authors for their detailed response and explanations.
>
> **1.** I look forward to seeing the strengthened theorem statement. I would be appreciate it if it can be provided before the end of the discussion period.
>
> **2.** I appreciate the informal explanation of the difference between CoT and autoregressive decoding, but my question was specifically about the formalization of CoT given in the paper. You say "we defined CoT within the context of SSMs by formalizing how the model recursively generates additional tokens based on its previous outputs and inputs", which is in agreement with my understanding of your definition. However, to me, that definition seems to be an accurate formalization of autoregressive decoding without incorporating the reasoning aspects of CoT.
>
> **3.** Making the changes to replace $\mathsf{L}$ with $\mathsf{FL}$ sounds good to me.
>
> I'm looking forward to the authors' responses to my questions 2-6 regarding Theorem 4.

---

> ### Author Response · Authors · 2024-11-28
> **Response to Reviewer yPps [Part II]**
>
> Dear Reviewer yPps,
>
> 1. We are very grateful for your insightful review. Your feedback has been instrumental in strengthening and changing Theorem 4. We have addressed all of your questions (points 2 to 6) by improving Theorem 4 and adding relevant explanations in the text following Theorem 3 and before the Experiments section. These updates are included in the revised manuscript.
>
> 2. You are correct in noting that we do not utilize external assistance in the "reasoning" process. This clarification aligns with the point raised by Reviewer BvEu, who initiated the discussion regarding AlphaProof-based external systems. If our paper is accepted, we will include a few additional sentences in the camera-ready version to further clarify and eliminate any ambiguity on this matter.
>
> 3. Thank you once again for your valuable observation. We did not put assumptions on A_t, B_t, C_t, and D_t since we now changed Theorem 4. Log assumptions had to be initially made, as in [1] on Page 6, Lemma 4.1.
>
> [1] The Illusion of State in State-Space Models; William Merrill, Jackson Petty, Ashish Sabharwal

---

> ### Comment · Reviewer_yPps · 2024-11-30
> **Response to authors' rebuttal Part II**
>
> ### 1
>
> I appreciate the authors' update to Theorem 4. The update resolves Question 2 that I had, and also renders my Questions 3-5 obsolete since the entire theorem statement was changed.
>
> Maybe putting the original Theorem 4 in the Appendix would be nice, and you can state in the main text "Under the log-precision assumption from Merrill et al. 2024, we prove Appendix Theorem ... that shows .... Here, we show using the more realistic assumption that precision doesn't depend on input sequence length that ..."
>
> The concern in my Question 6 remains. To quote one sentence from my original question:
> > My computer has finite memory, so does that mean it can only decide regular languages?
>
> The new Theorem 4 is identical in spirit to a theorem (call it Theorem 4') that says "A computer with finite memory is fundamentally limited to computations that can be performed by an FSM." And applying the takeaway in lines 426-431 to Theorem 4' will look like this:
> > These limitations are significant because they highlight the boundaries of what computers can achieve in
> practical settings. Regarding practical considerations, since real-world implementations of computers
> operate on hardware with finite memory and finite precision arithmetic, these theoretical limitations
> directly apply to computers used in actual applications. Therefore, when designing systems for tasks
> that require processing beyond regular languages, it becomes clear that computers
> may not suffice, and alternative architectures or computational mechanisms need to be considered to
> overcome these inherent constraints.
>
> which is the incorrect takeaway. The correct takeaway from Theorem 4' is that, although real computers operate with finite memory, the aptest theoretical model is one with infinite memory (i.e., a Turing machine). Similarly, I believe that the correct takeaway from the new Theorem 4 is that, although real SSMs operate with finite precision, the aptest theoretical model is one with infinite precision.
>
> ### 2
>
> Although the added paragraph in the paper does not directly address my question, I accept the formalization of CoT purely based on the fact that it was based on prior work. (However, I maintain that the formalization better captures the idea of autoregressive decoding and not of CoT.)
>
> ### 3
>
> Your definition of SSMs allows the matrices $A_t, B_t, C_t, D_t$ to depend on $t$, but your proof of Theorem 4 assumes fixed $A, B, C, D$. I would recommend explicitly stating this assumption in the theorem statement.
>
> ### About Theorem 1
>
> I appreciate the updated theorem and believe it is stronger than the previous one, although I have not yet checked the details of the proof.
>
> ### Summary
>
> I have revised my Soundness Score to 2 and Overall Score to 5 (Weak Reject).
> * _Explanation for not higher score:_ As elaborated above, my concern raised in Question 6 remains unresolved.
> * _Explanation for increase in score:_ Question 2 was resolved with the more realistic model of fixed precision. Also, although I would still recommend rejection based on my unresolved Question 6 alone, my objection is reduced by the fact that the authors are following the finite-precision assumption of previous work.

---

> > ### Author Response · Authors · 2024-12-01
> > **Response to Reviewer yPps [01.12.2024]**
> >
> > Dear Reviewer yPps,
> >
> > Thank you for your thoughtful and detailed feedback on our paper. We greatly appreciate the time and effort you have invested in reviewing our work, and your insights have been invaluable in improving the quality and clarity of our manuscript. We are also grateful for your updated score.
> >
> > ---
> >
> > **1. Inclusion of the Original Theorem 4:**
> >
> > We agree with your suggestion to include the original Theorem 4 in the appendix. We will add it as **Theorem 5** in the revised manuscript. In this appendix section, we will state the log-precision assumption given by Merrill et al. (2024). This will provide readers with a comprehensive understanding of the different assumptions under which our results hold and how they relate to prior work.
> >
> > ---
> >
> > **2. Addressing the Concern in Question 6:**
> >
> > We understand your concern regarding the practical implications of our result in Theorem 4. You mentioned that while physical computers have finite memory, we typically model them as Turing machines with infinite memory to capture their computational capabilities. Similarly, you suggest that although real SSMs operate with finite precision, the apt theoretical model might be one with infinite precision.
> >
> > Our intention with Theorem 4 was to highlight the theoretical limitations of SSMs when operating under practical constraints of finite precision and fixed hidden dimensions. We acknowledge that, in theory, allowing infinite precision could enable SSMs to perform more complex computations. However, in practice, neural network models, including SSMs, are implemented with finite precision due to hardware limitations.
> >
> > By demonstrating that SSMs with finite precision are computationally equivalent to finite-state machines, we aim to underscore the inherent limitations faced by these models in practical settings. This result emphasizes that, under realistic conditions, SSMs may not suffice for tasks requiring computational power beyond regular languages.
> >
> > We agree that this perspective aligns with understanding the limitations of real-world computers, which, despite having finite memory, can simulate Turing machines for practical purposes. However, finite memory limits the size and complexity of computations that can be performed in practice.
> >
> > To address your concern and provide a balanced view, we will revise the discussion following Theorem 4 to clarify that while our result highlights theoretical limitations under finite precision, it does not preclude the possibility of SSMs approximating more complex computations in practice. We will emphasize that our findings serve as a foundation for understanding the computational boundaries of SSMs and motivate the exploration of the capabilities under infinite-precision, along with architectures with improved capabilities.
> >
> > ---
> >
> > **3. Explicit Assumption on Matrices in Theorem 4:**
> >
> > Thank you for pointing out the need to explicitly state the assumption regarding the matrices A, B, C, and D in Theorem 4. We will revise the theorem statement to clearly specify that these matrices are fixed and do not depend on the input sequence.
> >
> > ---
> >
> > **4. Formalization of Chain-of-Thought (CoT):**
> >
> > We appreciate your acceptance of our formalization of CoT based on prior work, even though you maintain reservations about its distinction from autoregressive decoding. We will further refine our explanation in the manuscript to better highlight the aspects of CoT that involve explicit reasoning steps, distinguishing it more clearly from standard autoregressive decoding.
> >
> > ---
> >
> > **5. Appreciation for the Updated Theorem 1:**
> >
> > We are pleased you find the updated Theorem 1 stronger and more satisfactory. Your insights have greatly improved the rigor and clarity of our theoretical results.
> >
> > ---
> >
> > Once again, we sincerely thank you for your detailed and constructive review. Your feedback has significantly helped us improve our paper, and we are grateful for your willingness to engage deeply with our work. We have incorporated most of your suggestions into the revised manuscript and will incorporate others, and we believe these changes have strengthened the overall quality of our submission.

---

> > > ### Comment · Reviewer_yPps · 2024-12-03
> > >
> > > ### 2
> > >
> > > I would very much appreciate explicitly pointing out that Theorem 4 applies only to *finite precision* SSMs, and would highly recommend adding the qualification every time Theorem 4 is referenced (e.g. lines 506 to 510 in the conclusion of the updated paper). I would also appreciate offering a more balanced viewpoint in e.g. lines 426-431 of the updated paper.
> > >
> > > To maximize transparency to the reader, I would recommend pointing out the analogy between Theorem 4 and the statement that real computers are limited to FSMs because of their finite memory. It may be worth mentioning that, as a result, the practical implications of the theorem statement are more relevant for heavily quantized SSMs. (Perhaps to partially justify the finite-precision model you may also make a connection to existing work suggesting a real-valued parameter really acts as though it's a small finite number of bits.) And then you may suggest future work that addresses the concern by assuming infinite precision.
> > >
> > > In summary, I believe the current new additions in the paper exaggerate the practical implications of Theorem 4, and should be significantly modified/reduced for a more accurate portrayal. **My score increase assumes that the authors will implement my recommendations here.**

---

> ### Comment · Reviewer_yPps · 2024-12-03
> **Final summary**
>
> I very much thank the authors for engaging in productive discussions about their paper. From the start to the end of the discussion period, I increased my score from 3 (reject) to 6 (borderline accept).
>
> A few of my questions and concerns have been addressed, and the main concern remaining pertains to the practical implications of Theorem 4. Assuming that the authors will address my [current concerns](https://openreview.net/forum?id=DhdqML3FdM&noteId=qems7DrZHo) that version 2 of the manuscript oversells the implications of the result, I recommend weak acceptance of the paper.
>
> Sincerely,
>
> Reviewer yPps

---

> > ### Author Response · Authors · 2024-12-03
> > **Response to Reviewer yPps [03.12.2024]**
> >
> > Dear Reviewer yPps,
> >
> > Thank you for your thoughtful feedback and for engaging in productive discussions about our paper. We appreciate your suggestions regarding Theorem 4 and will incorporate them into the revised manuscript. Specifically, we will explicitly state that Theorem 4 applies only to finite precision SSMs wherever it is referenced, adjust the discussion to offer a more balanced viewpoint, and include the analogy to real computers and FSMs to improve transparency for the reader.
> >
> > Thank you again for your valuable input, which has been instrumental/critical in improving our work.

---

### Author Response · Authors · 2024-11-28
**[Final comment] Manuscript is updated**

Dear Reviewers,

We would like to express our gratitude for your insightful and constructive feedback. In response to your comments, we have thoroughly revised our manuscript to address all concerns and improve the overall quality of our work. Below, we outline the key changes made:

- **Theorem 1 Change**: Added the prompt order to Theorem 1 to provide a clearer understanding of its application.
- **Theorem 4 Revision**: Replaced Theorem 4 to better align with the revised framework and improve its robustness.
- **External Engines Integration**: Included citations to AlphaProof-based systems and other external engines to contextualize our work within existing technologies.
- **AGI Implications**: We moved the discussion on AGI implications to the conclusion section and removed it from the related work to highlight its significance better.
- **Appendix Expansion**: A comprehensive background on communication complexity was added to the appendix to provide additional context and support for our findings.

All revisions in the updated manuscript are highlighted in blue for your convenience. Although the discussion deadline has passed, we remain committed to refining our work. We are open to addressing any remaining typos or minor issues with the camera-ready version if we receive the acceptance. We will re-read the manuscript to correct any small misalignments and ensure clarity and coherence.

Once again, we sincerely thank you for your valuable feedback, which has been instrumental in improving our paper. We hope the revisions satisfactorily address your concerns and that you will consider updating your evaluations/ratings accordingly.

Best regards,

authors of "Limits of Deep Learning: Sequence Modeling through the Lens of Complexity Theory."

---

### Meta-Review · Area_Chair_gtjJ · 2024-12-19

**Metareview:**

This paper empirically and theoretically analyzes the computational limitations of Structured State Space Models (SSMs). The authors introduce three theorems: the inability of SSMs to compose functions, exponential scaling of compute when performing chain-of-thought, and an inability to solve NL-complete problems (unless L=NL). These theorems are backed up by empirical evidence in several different reasoning tasks.

Most reviewers agreed that the paper is original: it echos findings for transformers but applies them to the SSM architecture and yields new theoretical insights which are likely to be very valuable to the SSM community. These theoretical insights are further backed up by experiments demonstrating the limitations of function composition in SSMs. Overall, reviewers generally agreed that the paper was well-structured, with proofs being clear for each theorem.

As is common for theoretical papers of this kind, and as noted by several reviewers, it is not clear how large the potential gap is between the theoretical results presented here and practical applications. For example, in practice, methods such as search and self-correction are often used in combination with SSMs to handle complex problems, and as a result it may be possible to mitigate some of the challenges of SSMs by relying on these. However, the authors now note this directly in the revised manuscript, and reviewers agree that understanding the limitations of SSMs architecturally (without such additions) is still very valuable.

All reviewers agree that this paper is worth accepting for the reasons above, and I similarly recommend acceptance.

**Additional Comments On Reviewer Discussion:**

The authors and reviewers engaged in very productive discussion during the rebuttal period which led to significant improvements in the paper, and the raising of overall scores (in some cases by multiple points).

Theorem 4 substantially improved based on the feedback from several reviewers, and the authors added a significantly improved conclusion section which also touches on limitations.

---

### Decision · Program_Chairs · 2025-01-22

Accept (Poster)